# Multi-fidelity Monte Carlo:
# a pseudo-marginal approach

**Diana Cai**
Department of Computer Science
Princeton University
dcai@cs.princeton.edu

**Ryan P. Adams**
Department of Computer Science
Princeton University
rpa@princeton.edu

## Abstract

Markov chain Monte Carlo (MCMC) is an established approach for uncertainty quantification and propagation in scientific applications. A key challenge in applying MCMC to scientific domains is computation: the target density of interest is often a function of expensive computations, such as a high-fidelity physical simulation, an intractable integral, or a slowly-converging iterative algorithm. Thus, using an MCMC algorithms with an expensive target density becomes impractical, as these expensive computations need to be evaluated at each iteration of the algorithm. In practice, these computations often approximated via a cheaper, low-fidelity computation, leading to bias in the resulting target density. Multi-fidelity MCMC algorithms combine models of varying fidelities in order to obtain an approximate target density with lower computational cost. In this paper, we describe a class of asymptotically exact multi-fidelity MCMC algorithms for the setting where a sequence of models of increasing fidelity can be computed that approximates the expensive target density of interest. We take a pseudo-marginal MCMC approach for multi-fidelity inference that utilizes a cheaper, randomized-fidelity unbiased estimator of the target fidelity constructed via random truncation of a telescoping series of the low-fidelity sequence of models. Finally, we discuss and evaluate the proposed multi-fidelity MCMC approach on several applications, including log-Gaussian Cox process modeling, Bayesian ODE system identification, PDE-constrained optimization, and Gaussian process parameter inference.

## 1   Introduction

Simulation and computational modeling play a key role in science, engineering, economics, and many other areas. When these models are high-quality and accurate, they are important for scientific discovery, design, and data-driven decision making. However, the ability to accurately model complex physical phenomena often comes with a significant cost—many models involve expensive computations that then need to be evaluated repeatedly in, for instance, a sampling or optimization algorithm. Examples of model classes with expensive computations include intractable integrals or sums, expensive quantum simulations [43], expensive numerical simulations arising from partial differential equations (PDEs) [38] and large systems of ordinary equations (ODEs).

In many situations, one has the ability to trade off computational cost against *fidelity* or accuracy in the result. Such a tradeoff might arise from the choice of discretization or the number of basis functions when solving a PDE, or the number of quadrature points when estimating an integral. It is often possible to leverage lower-fidelity models to help accelerate high-quality solutions, e.g., by using multigrid methods [23] for spatial discretizations. More generally, *multi-fidelity* methods combine multiple models of varying cost and fidelity to accelerate computational algorithms and have been applied to solving inverse problems [11, 24, 38], trust region optimization [1, 4, 15, 31, 39], Bayesian optimization [8, 21, 27, 28, 41, 45], Bayesian quadrature [17, 46], and sequential learning [22, 35].

36th Conference on Neural Information Processing Systems (NeurIPS 2022).


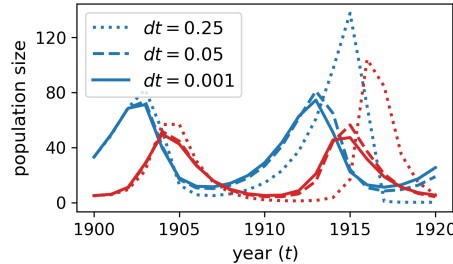

(a) Trapezoid rule with $2k$ trapezoids ($k = 1, 2, 3$)    (b) Lokta-Volterra ODE solutions, $dt = 1/k$

Figure 1: Examples of low-fidelity sequences of models. **(a)** Sequence trapezoid quadrature estimates $I_k$, where $I_k$ is the trapzoid rule with $2k$ trapezoids. **(b)** Lokta-Volterra ODE solutions for prey $u(t)$ (blue) and predator $v(t)$ (red) using Euler's method with step size $dt$.

One critically important tool for scientific and engineering computation is Markov chain Monte Carlo (MCMC), which is widely used for uncertainty quantification, optimization, and integration. MCMC methods are recipes for constructing a Markov chain with some desired target distribution as the limiting distribution. Pseudo-random numbers are used to simulate transitions of the Markov chain in order to produce samples from the target distribution. However, MCMC often becomes impractical for high-fidelity models, where a single step of the Markov chain may, for instance, involve a numerical simulation that takes hours or days to complete. Multi-fidelity methods for MCMC focus on constructing Markov chain transition operators that are sometimes able to use inexpensive low-fidelity evaluations instead of expensive high-fidelity evaluations. The goal is to increase the effective number of samples generated by the algorithm, given a constrained computational budget. A large focus of the multi-fidelity MCMC literature is on two-stage Metropolis-Hastings (M-H) methods [10, 14], which use a single low-fidelity model for early rejection of a proposed sample, thereby often short-circuiting the evaluation of the expensive, high-fidelity model.

However, there are several limitations of two-stage multi-fidelity Monte Carlo. First, in many applications, a *hierarchy* of cheaper, low-fidelity models is available; for instance, in the case of integration, $k$-point quadrature estimates form a hierarchy of low-fidelity models, and in the case of a PDE, varying the discretization. Thus, the two-stage approach does not fully utilize the availability of a hierarchy of fidelities and may be more suitable for settings where the high- and low-fidelity models are not hierarchically related, e.g., semi-empirical methods vs. Hartree-Fock in computational chemistry. In addition, in such applications, there is often a limiting model of interest, such as a continuous function that the low-fidelity discretizations approximate. Two-stage MCMC does not asymptotically sample from this limiting target density and will at best sample from an approximation of the biased, high-fidelity posterior. Finally, the two-stage method is unnatural to generalize to more sophisticated MCMC algorithms such as slice sampling and Hamiltonian Monte Carlo (HMC).

We propose a class of multi-fidelity MCMC methods designed for applications with a hierarchy of low-fidelity models available. More specifically, we assume access to a sequence of low-fidelity models that converge to a "perfect-fidelity" model in the limit. Within an MCMC algorithm, we can approximate the perfect-fidelity target density with an unbiased estimator constructed from a randomized truncation of the infinite telescoping series of low-fidelity target densities. This class of multi-fidelity MCMC is an example of a pseudo-marginal MCMC (PM-MCMC) algorithm—the unbiased estimator essentially guarantees that the algorithm is asymptotically exact in that the limiting distribution recovers the perfect-fidelity target distribution as its marginal distribution. Our approach introduces the fidelity of a model as an auxiliary random variable that is evolved separately from the target variable according to its own conditional target distribution; this technique can be used in conjunction with any suitable MCMC update that leaves the conditional update for the target variable of interest invariant, such as M-H, slice sampling, elliptical slice sampling, or Hamiltonian Monte Carlo. We apply the pseudo-marginal multi-fidelity MCMC approach to several problems, including log-Gaussian Cox process modeling, Bayesian ODE system identification, PDE-constrained optimization, and Gaussian process parameter inference.

**Related work.** Multi-fidelity MCMC methods are commonly applied in a two-stage procedure, where the goal is to reduce the computational cost of using a single expensive high-fidelity model by

using a cheap low-fidelity model as a low-pass filter for a delayed acceptance/rejection algorithm [10, 11, 14]; see Peherstorfer et al. [36] for a survey. Higdon et al. [24] propose coupling a high-fidelity Markov chain with a low-fidelity Markov chain via a product chain. In constrast, our approach aims to sample from a "perfect-fidelity" target density while reducing computational cost; two-stage MCMC algorithms result in biased estimates with respect to this target density. A related class of methods is multilevel Monte Carlo [13, 18, 19, 44], which uses a hierarchy of multi-fidelity models for Monte Carlo estimation by expressing the expectation of a high-fidelity model as a telescoping sum of low-fidelity models. Dodwell et al. [13] use the M-H algorithm to form the multilevel Monte Carlo estimates, simulating from a separate Markov chain for each level of the telescoping sum. In practice multilevel Monte carlo requires choosing a finite number of fidelities, inducing bias in the estimator with respect to the (limiting) perfect-fidelity model. In contrast, our method uses a randomized fidelity within a single Markov chain with the perfect-fidelity model as the target.

Our approach applies pseudo-marginal MCMC to multi-fidelity problems. There is a rich literature developing pseudo-marginal MCMC methods [2, 6] for so-called "doubly-intractable" likelihoods, which are likelihoods that are intractable to evaluate. Several approaches in the pseudo-marginal MCMC literature are particular relevant to our work. The first are the PM-MCMC methods introduced by Lyne et al. [30], which describes a class of pseudo-marginal M-H methods that use Russian roulette estimators to obtain unbiased estimators of the likelihood. However, this method samples the variable of interest jointly with the auxillary randomness, which often leads to sticking.

Alternatively, several methods have considered sampling the randomness separately. The idea of clamping random numbers is explored in depth by Andrieu et al. [3] and Murray and Graham [33]; the latter applies to this pseudo-marginal slice sampling. In particular, our approach applies these ideas to the specific setting of multi-fidelity models, where the random fidelity is treated as an auxillary variable. Finally, while our approach applies to doubly-intractable problems, we are also motivated by a larger class of multi-fidelity problems studied in the computational sciences that may not even be inference problems, such as quantum simulations and PDE-constrained optimization.

## 2   Multi-fidelity MCMC

Monte Carlo methods approximate integrals and sums that can be expressed an expectation:

$$\mathbb{E}_\pi(h(\theta)) = \int h(\theta)\, \pi(\theta)\, d\theta \approx \frac{1}{T} \sum_{t=1}^{T} h(\theta^{(t)}), \qquad \text{where} \quad \theta^{(t)} \sim \pi, \tag{1}$$

and where $\pi : \Theta \to \mathbb{R}_+$ is the *target density* that may only be known up to a constant, $h(\theta)$ is a function of interest, and $\{\theta^{(t)}\}_{t=1}^{T}$ are samples from $\pi$. Markov chain Monte Carlo methods are then used to generate samples $\theta^{(t)}$ from $\pi$ by simulating from a Markov chain with target $\pi$.

In many settings, pointwise evaluations of the target function $\pi(\theta)$ are expensive or even intractable; from here on we will assume that the goal is to compute statistics of a quantity of interest $h(\theta)$ with respect to a *perfect-fidelity* target density $\pi_\infty(\theta)$. In practice, the estimate in Equation (1) is instead estimated using a cheaper, low-fidelity density $\pi_k(\theta)$, where $k \in \mathbb{N} := \{1, 2, \ldots\}$. In particular, we consider settings where there is a *sequence* of low-fidelity densities available that converge to the target, i.e., $\pi_k(\theta) \xrightarrow{k \to \infty} \pi_\infty(\theta)$. We assume that as $k$ increases, the model becomes higher in fidelity (with respect to $\pi_\infty$) but more costly to evaluate, increasing in expense super-linearly with $k$.

For instance, $\pi_\infty$ could represent a target density that depends on an intractable integral, the solution of a PDE, the solution of a large system of ODEs, the solution of a large system of linear equations, or the minimizer of a function. Thus, a typical evaluation of $\pi_\infty$ requires an approximation at a fidelity $k$ with a tolerable level of bias for a given computational budget. Here increasing $k$ could correspond to finer discretizations of differential equations, increasing numbers of quadrature points, or performing a larger number of iterations in a linear solver or optimization routine.

In the multi-fidelity setting, the goal is to combine several models of varying fidelity within an MCMC algorithm to reduce the computational cost of estimating Equation (1). In this paper, we describe a class of MCMC algorithms that leverages the sequence of low-fidelity models $\pi_k$. Our strategy for multi-fidelity MCMC (MF-MCMC) will be to construct an unbiased estimator of $\pi_\infty(\theta)$ using random choices of the fidelity $K$ and then to include $K$ in the Markov chain as an auxiliary variable. By carefully constructing such a Markov chain, it will be possible to asymptotically estimate the

functional in Equation (1) as though the samples were taken from the perfect-fidelity model; each step of the Markov chain will nevertheless only require a finite amount of computation. Finally, our approach allows us to essentially plug in any valid MCMC algorithm, and we apply this strategy to develop multi-fidelity variants of a number of MCMC algorithms, such as M-H and slice sampling.

## 2.1 Pseudo-marginal MCMC for the multi-fidelity setting

Pseudo-marginal MCMC [2, 6] is a class of auxillary-variable MCMC algorithms that replaces the target density $\pi(\theta)$ with an estimator $\hat{\pi}(\theta)$ that is a function of a random variable. If the estimator is nonnegative and unbiased, i.e., for all $\theta \in \Theta$, $\hat{\pi}(\theta) \geq 0$ and $\mathbb{E}[\hat{\pi}(\theta)] = \pi(\theta)$, then MCMC transitions that use the estimator still have $\pi(\theta)$ as their invariant distribution. This property is sometimes referred to as "exact-approximate" MCMC as the transitions are approximate but the limiting distribution is exact. Estimators can be constructed from a variety of methods, including particle filtering [2]; our approach will use randomized series truncations, which has been consider in pseudo-marginal MCMC methods such as Lyne et al. [30], Georgoulas et al. [16], and Biron-Lattes et al. [7].

We now apply the pseudo-marginal approach to the multi-fidelity setting. Here the target density estimator arises from a random choice of the fidelity $K \in \mathbb{N}$ that is governed by a distribution $\mu$ on $\mathbb{N}$. We denote the estimator using $\hat{\pi}_K(\theta)$ to make the dependence on the random fidelity $K$ explicit. The estimator is constructed such that it is unbiased with respect to $\mu$, i.e.,

$$\sum_{k=1}^{\infty} \mu(k)\hat{\pi}_k(\theta) = \pi_{\infty}(\theta). \tag{2}$$

The distribution $\mu$ is also constructed by the user: ideally, the estimator $\hat{\pi}_K(\theta)$ will prefer smaller values of $K$ while having sufficiently low variance as to allow the Markov chain to mix effectively. Thus the simulations can be run at inexpensive low-fidelities, while the estimates will be as though the perfect-fidelity model were being used.

The standard pseudo-marginal MCMC approach is to construct a Markov chain that has the following joint density as its stationary distribution:

$$\pi(\theta, K) = \mu(K)\hat{\pi}_K(\theta). \tag{3}$$

Observe that while Equation (3) does not depend on the perfect-fidelity target density $\pi_{\infty}$, it returns the desired marginal $\pi_{\infty}$ via Equation (2). As a concrete example, a pseudo-marginal M-H algorithm generates a new state $\theta'$ and fidelity $K'$ jointly using $q(\theta'; \theta)$ as the proposal for $\theta'$, $q(K'; K) = \mu(K')$ as the proposal distribution for the fidelity, and accepts/rejects the state according to

$$a = \frac{\pi(\theta', K')q(\theta; \theta')q(K; K')}{\pi(\theta, K)q(\theta'; \theta)q(K'; K)} = \frac{\hat{\pi}_{K'}(\theta')q(\theta; \theta')}{\hat{\pi}_K(\theta)q(\theta'; \theta)}, \tag{4}$$

where the equality holds since the distribution terms for $K$ and $K'$ cancel. Note that the right-hand side of Equation (4) is the standard M-H ratio but that the target density $\pi$ is replaced with the estimator $\hat{\pi}_K$.

However, standard pseudo-marginal MCMC using joint proposals of the state and fidelity can "get stuck" when the estimator is noisy and fail to accept new states. Thus, we apply the approach in Murray and Graham [33] that augments the Markov chain to include the randomness of the estimator via a separate update; here the randomness of the estimator arises from the fidelity $K$. Concretely, we construct a Markov chain that simulates from Equation (3) by alternating sampling between the conditional target densities $\pi(K|\theta)$ and $\pi(\theta|K)$ (steps 5 and 6 of Algorithm 1, respectively). We refer to this strategy as *multi-fidelity MCMC* (MF-MCMC), since by conditioning on $K = k$, the update for the state $\theta$ becomes a standard deterministic update applied to a low-fidelity model $\hat{\pi}_k(\theta)$, and any appropriate MCMC update can be used here, making it straightforward to use complex MCMC methods, such as slice sampling and HMC. Similarly, any suitable MCMC update for the fidelity $K$ can be used using the conditional target $\pi(K|\theta)$.

Many techniques can be used to construct an unbiased estimator of $\pi_{\infty}$ with randomness $K$; we describe a general approach in the next section. However, it is generally difficult to guarantee the estimator is nonnegative, as required by pseudo-marginal MCMC. One technique considered by Lin et al. [29] and Lyne et al. [30] is to instead sample from the target distribution induced by the absolute value of the estimator and applying a sign-correction to the final Monte Carlo estimate in

Equation (1), an approach borrowed from the quantum Monte Carlo literature where it is necessary for modeling fermionic particles. This approach has been applied to the M-H algorithm, but we note that this general approach can be applied much more broadly, as we do in this work.

In problems where the estimator may be negative, we sample from the conditional target distributions using the absolute value of the estimator $|\hat{\pi}_K(\theta)|$, and we denote these conditionals with $\tilde{\pi}(K \mid \theta) \propto \mu(K)|\hat{\pi}_K(\theta)|$ and $\tilde{\pi}(\theta|K = k) \propto |\hat{\pi}_k(\theta)|$. The estimate in Equation (1) is then corrected using the signs $\sigma(\theta, k)$ of evaluations of $\hat{\pi}_k(\theta)$,

$$\int h(\theta)\, \pi(\theta)\, d\theta \approx \frac{\sum_{t=1}^{T} h(\theta^{(t)})\sigma(\theta^{(t)}, K^{(t)})}{\sum_{t=1}^{T} \sigma(\theta^{(t)}, K^{(t)})} =: \hat{I}_T, \tag{5}$$

where $\{(\theta^{(t)}, K^{(t)})\}_{t=1}^{T}$ are the sampled values from the joint distribution $\tilde{\pi}(\theta, K) \propto |\hat{\pi}_K(\theta)|\mu(K)$.

Importantly, the sign-corrected estimate still asymptotically leads to the desired estimate of the functional of interest. Let $\sigma(\theta, k)$ denote the sign of the estimator such that $\hat{\pi}_k(\theta) = \sigma(\theta, k)|\hat{\pi}_k(\theta)|$. The estimator $\hat{I}_T$ in Equation (5) is formed using a Monte Carlo estimate of the functional after expanding it into its joint distribution, i.e.,

$$\int h(\theta)\pi_\infty(\theta)d\theta = \int \sum_{k=1}^{\infty} h(\theta)\hat{\pi}_k(\theta)\mu(k)d\theta = \frac{\int \sum_{k=1}^{\infty} h(\theta)\sigma(\theta, k)\tilde{\pi}(\theta, k)d\theta}{\int \sum_{k=1}^{\infty} \sigma(\theta, k)\tilde{\pi}(\theta, k)d\theta}. \tag{6}$$

The full multi-fidelity MCMC algorithm with sign correction summarized in Algorithm 1. We note that while the Markov chain no longer converges to a target with the marginal $\pi_\infty$, the final estimate after sign-correction—which is the downstream goal of interest—converges to the quantity of interest due to Equation (6). While this may seem limiting if one is interested in the posterior itself, useful unbiased posterior summaries may be still be obtained via the functional, such as the posterior mean, variance, quantiles, and histograms that may be used to visualize marginal distributions.

## 3 Unbiased low-fidelity estimators via randomized truncations

In this section, we discuss how to construct an unbiased estimator of $\pi_\infty(\theta)$, given a sequence of low-fidelity likelihoods with the property $\pi_k(\theta) \to \pi_\infty(\theta)$ as $k \to \infty$. This estimator has the property that it requires a finite amount of computation with probability one, and it also has a tunable amount of expected computation per estimate, i.e., it uses low-fidelity density evaluations to estimate the perfect-fidelity target density. The central idea of this estimator has been used for decades, going back to John von Neumann and Stanislaw Ulam. More recently it has found use in applications of inference and optimization in related work such as Glynn and Rhee [20], Lyne et al. [30], Beatson and Adams [5], and Jacob et al. [26].

First note that we can express the perfect-fidelity model as a telescoping sum of low-fidelity models: let $\pi_0(\theta) = 0$ and write

$$\pi_\infty(\theta) = \sum_{k=1}^{\infty} \pi_k(\theta) - \pi_{k-1}(\theta). \tag{7}$$

The estimator $\hat{\pi}_K$ is then constructed by taking a random truncation $K \sim \mu$ of the infinite telescoping series. The sampled terms in the sum are then reweighted to ensure the estimator remains unbiased:

$$\hat{\pi}_K(\theta) = \sum_{k=1}^{K} w_{k,K}(\pi_k(\theta) - \pi_{k-1}(\theta)). \tag{8}$$

Two approaches are commonly used to ensure that the resulting estimator is unbiased: weighted single-term estimators and Russian roulette estimators. The single-term estimator [30] is constructed by importance sampling a term from the series in Equation (7): the truncation level is drawn as $K \sim \mu$, and the $K$th term is used to form the estimate

$$\hat{\pi}_K(\theta) = \mu(K)^{-1}(\pi_K(\theta) - \pi_{K-1}(\theta)). \tag{9}$$

Thus, the weight in Equation (8) is $W_{k,K} = \mu(K)^{-1}\mathbb{1}(K = k)$. In the Russian roulette estimator, the remaining terms in the estimator are reweighted by their survival probabilities,

---

**Algorithm 1** Multi-fidelity Monte Carlo with sign-correction

---

1: **Input**: Initial state $\theta$ and fidelity $K$, truncation distribution $\mu$
2: **for** $t = 1, \ldots, T$ **do**
3:     Given current $K$ and $\theta$, form estimator $\hat{\pi}_K(\theta) = \sum_{k=1}^{K} w_{k,K}(\pi_k(\theta) - \pi_{k-1}(\theta))$
4:     Save sign $\sigma(\theta, K) = \text{sign}(\hat{\pi}_K(\theta))$
5:     Update fidelity $K$ leaving invariant the target conditional

$$\tilde{\pi}(K|\theta) \propto \mu(K)|\hat{\pi}_K(\theta)|$$

6:     Update state $\theta$ leaving invariant the target conditional

$$\tilde{\pi}(\theta|K = k) \propto |\hat{\pi}_k(\theta)|$$

7: **end for**
8: **Output**: Samples $\{(\theta^{(t)}, K^{(t)})\}$ and estimate $\hat{I}_T = \left(\sum_{t=1}^{T} \sigma^{(t)} h(\theta^{(t)})\right) / \left(\sum_{t=1}^{T} \sigma^{(t)}\right)$

---

i.e., $W_{k,K} = (1 - \sum_{k'=1}^{k-1} \mu(k'))^{-1} \mathbb{1}(K \geq k)$. The distribution $\mu$ controls the number of terms in the estimator, and a good proposal distribution should match the tails of the sequence of low-fidelity densities [5, 30, 37].

The ability to use cheaper models is a key feature of multi-fidelity inference, and the low-fidelity estimator provides a means to reduce the computational cost of multi-fidelity Monte Carlo. However, these estimators are an example of a class of methods that explores a compute-variance tradeoff: computationally cheaper estimates leads to high variability. The resulting increase in variance slows down the convergence of the MCMC procedure and could lead to an overall less efficient method due to a reduced effective sample size.

## 4  Summary of the multi-fidelity MCMC recipe

Here we summarize the recipe for constructing a multi-fidelity Markov chain Monte Carlo algorithm.

First, identify a sequence of increasing-fidelity target densities with the property that their limit is the desired "perfect-fidelity" density. Low-fidelity densities should be cheap with the cost rapidly increasing within the sequence. In the context of Bayesian inference, it may be appropriate to focus the multi-fidelity aspects on the likelihood term and construct the target densities via, e.g., $\pi_k(\theta; \mathcal{D}) \propto \pi_0(\theta) L_k(\theta; \mathcal{D})$, where $\pi_0$ is the prior, $L_k$ is a low-fidelity likelihood, and $\mathcal{D}$ is the set of observations. This likelihood-based version is what we use in several of the experiments.

Next, introduce a truncation distribution $\mu$ on $\mathbb{N}$. This truncation distribution should be chosen to balance between expected cost and variance of the resulting estimator; our overall goal is to mostly use cheap low-fidelity densities, but high-variance estimators will presumably damage the mixing time and/or the asymptotic variance.

Initialize the Markov chain with a reasonable choice for $\theta$ and a draw of $K$ from the distribution $\mu$. Each step of the Markov chain simulation consists of an update to $\theta$ given $K$ and an update of $K$ given $\theta$. The update of $\theta$ given $K$ can be performed using any standard MCMC algorithm, e.g., M-H, slice sampling, or HMC, applied to the low-fidelity estimator. It is important to use the absolute value of the estimator and keep track of its sign. The update of $K$ given $\theta$ is also flexible, but it is reasonable to construct the update so that only a few $K$ are considered in each step, as each of those fidelities will need to be evaluated. By default, we consider a simple random walk on the positive integers for our experiments. After running a sufficient number of steps of the Markov chain, use the sign corrected-estimator in Equation (5) to compute the expectation of the function $h(\theta)$.

## 5  Experiments

In all experiments, we use a random-walk M-H update to sample from the conditional $K|\theta$, and truncation distribution $\mu(K) = \text{geometric}(K; \gamma_0)$. Additional experimental details are in Appendix F.

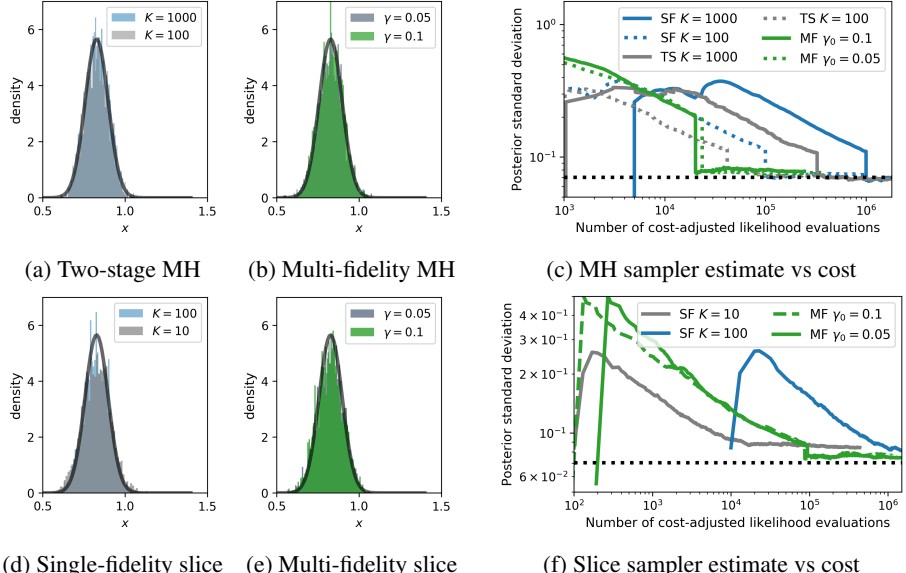

| (a) Two-stage MH | (b) Multi-fidelity MH | (c) MH sampler estimate vs cost |
|---|---|---|

| (d) Single-fidelity slice | (e) Multi-fidelity slice | (f) Slice sampler estimate vs cost |
|---|---|---|

Figure 2: Conjugate Gaussian model. **Left:** Histograms for M-H (a,b) and slice sampling (d,e). **Right:** Comparison of posterior standard deviation estimate vs computation for M-H (c) and slice sampling (d) methods. Average posterior mean computed over 4 different chains.

## 5.1 Toy conjugate Gaussian models

In order to understand the behavior of MF-MCMC on a simple example of Bayesian inference, we first examine an example where the computational cost of evaluating the sequence of low-fidelity likelihoods does not increase with $k$. Consider a perfect-fidelity likelihood $L_\infty(\theta) = \mathcal{N}(x; \theta, \sigma_\infty)$ and a low-fidelity likelihood $L_k(\theta) = \mathcal{N}(x; \theta, \sigma_k)$, where $\sigma_k^2 \to \sigma_\infty^2$. The prior is $\pi_0(\theta) = N(\theta|0, 1)$, and so a closed-form posterior density can be computed. Here we consider the sequence $\sigma_k^2 = 1 + 2/k^2$ and $\sigma_\infty^2 = 1$. In Figure 2 we compare the results of single-fidelity and multi-fidelity M-H and slice sampling as well as the two-stage M-H algorithm summarized in Appendix C.4. We consider 2 two-stage M-H settings with high and low fidelities of $\{k^{\text{HF}}, k^{\text{LF}}\} = \{1000, 10\}$ and $\{k^{\text{HF}}, k^{\text{LF}}\} = \{100, 5\}$. The histograms show the bias of each method after simulating 10,000 samples, and the solid gray curve denotes the exact posterior density. We also compute a measure of total cost and a running average of the estimate of the posterior standard deviation functional, where the dotted black line denotes the true value. The number of cost-adjusted likelihoods was computed by upweighting each likelihood evaluation by the fidelity. Here the multi-fidelity methods typically converge to a similar value as the single high-fidelity methods but in fewer cost-adjusted likelihood evaluations.

## 5.2 Log-Gaussian Cox processes

We examine an application of MF-MCMC to the log Gaussian Cox process (LGCP) model [32], where the perfect-fidelity model is a function of an integral and the lower-fidelity sequence of models arises from $k$-point quadrature estimates. Let $\log f \sim \text{GP}(0, \kappa_\ell)$, where $\kappa_\ell(x, x') = \exp\left(-\frac{1}{2\ell^2}\|x - x'\|_2^2\right)$ and where $\ell$ is a lengthscale hyperparameter. Consider an inhomogenous Poisson process on $\mathbb{X} \subseteq \mathbb{R}^D$ with intensity $\lambda(x) = e^{f(x)}$. Given a random set of points $\{X_n\}_{n=1}^N$, the perfect-fidelity likelihood is

$$L_\infty(f) = \exp\left(\int_{\mathbb{X}} (1 - e^{f(x)}) dx\right) \prod_{n=1}^N e^{f(X_n)}. \tag{10}$$

Typically, inference in the LGCP uses a grid-based approximation of Equation (10), where the points are binned into counts and modeled with a Poisson likelihood [12, 34, 42], resulting in a biased posterior. Because the likelihood depends on a high-dimensional latent Gaussian vector, we perform inference for $f$ using the elliptical slice sampling (ESS) algorithm (see Appendix C.3). We approximate the integral in Equation (10) with a trapezoidal quadrature rule $I_k$, where the number of quadrature points is a linear function of $k$.

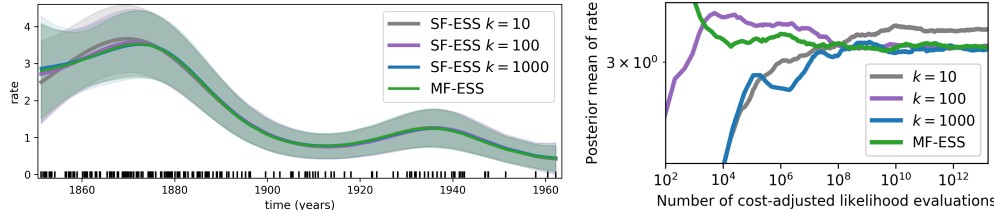

Figure 3: Coal mining disasters 1850–1963. **Left:** Posterior mean of the rate function at the observed data points. **Right:** Posterior mean of the rate function at $T = 1862$ vs computation.

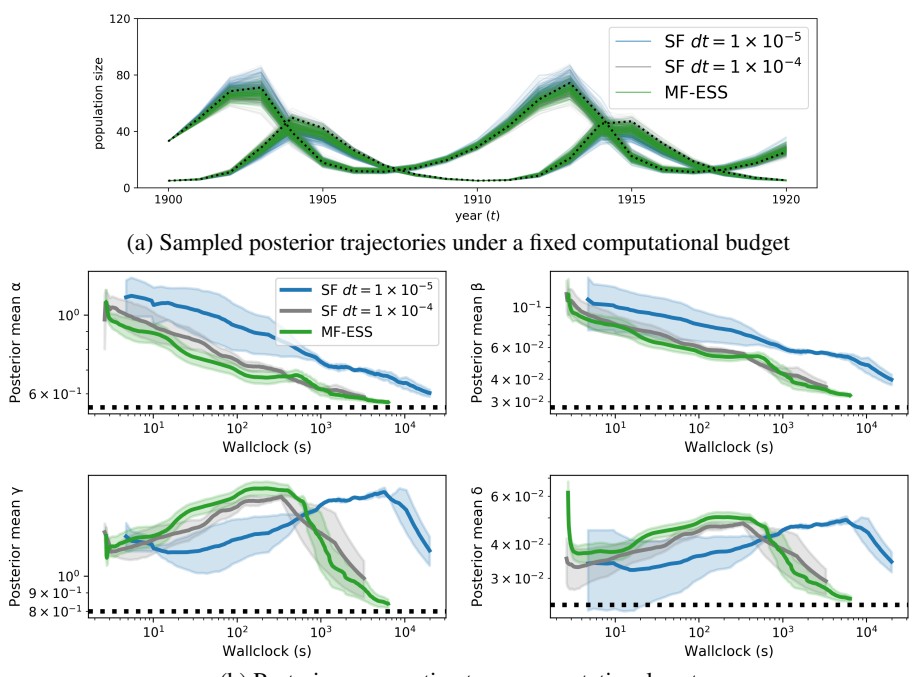

(a) Sampled posterior trajectories under a fixed computational budget

(b) Posterior mean estimate vs computational cost

Figure 4: Lokta-Volterra system parameter identification. The fidelity represents (a function of) the step size $dt$ of the ODE solver.

We apply multi-fidelity and single-fidelity ESS algorithms to a coal mining disasters data set (Carlin et al. [9]). The data contain the dates of 191 coal mine explosions that killed ten or more men in Britain between March 15, 1851 and March 22, 1962. Figure 3 (left) shows the estimated mean intensity and standard deviation on coal mining disasters data between one run of multi-fidelity ESS and single-fidelity ESS with $k = 10, 100, 1000$ quadrature points. In this plot, the high- ($k = 1000$) and multi-fidelity posterior mean and standard deviation estimates match well, and the bias in the lowest fidelity ($k = 10$) estimate is apparent. We also computed the cost-adjusted number of likelihood evaluations performed in each iteration of MF-ESS and SF-ESS. Figure 3 (right) shows the average estimated mean intensity at the time step $t = 1862$ on the three models against the average cost-adjusted number of likelihood evaluations per iteration. We observe that the multi-fidelity and high-fidelity estimates are close after many iterations of sampling, but that the multi-fidelity estimate converges with less computation.

## 5.3 Bayesian ODE system identification

We now apply the MF-MCMC approach to Bayesian system identification for the Lotka-Volterra ODE. Let $u(t) \geq 0$ represent the population size of the prey species at time $t$, and $v(t) \geq 0$ represent the population size of the predator species. The dynamics of the two species given parameters $\alpha, \beta, \gamma, \delta \geq 0$ are given by a pair of first-order ODEs:

$$\frac{d}{dt}u = (\alpha - \beta v)u = \alpha u - \beta uv, \qquad \frac{d}{dt}v = (-\gamma - \delta u)v = -\gamma v - \delta uv. \qquad (11)$$

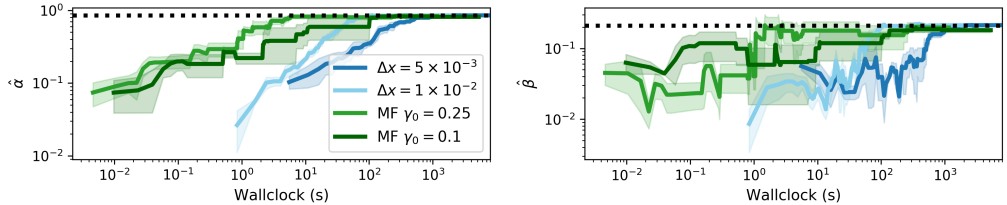

Figure 5: PDE-constrained optimization with a linear heat equation. Estimate for $\alpha$ (left) and $\beta$ (right) vs computation. The black dotted lines denote the true values of $\alpha, \beta$.

System identification solves the inverse problem by estimating the parameters of the ODE system $\theta = (\alpha, \beta, \gamma, \delta)$. Taking a Bayesian approach, we specify a noise model for the observed data and priors on the parameters, and we use MCMC to infer a distribution over the solution. For simplicity, we assume that the initial conditions are known and fix $\sigma = 0.25$.

Define $z_n := (u(t_n), v(t_n))$ and let $z_1(\theta), \dots, z_N(\theta)$ be the solutions to the Lotka-Volterra differential equations at times $t_1, \dots, t_N$ given the initial conditions and the system parameters $\theta = (\alpha, \beta, \gamma, \delta)$. Suppose we have observations arising from $\log(y_n) = \log(z_n) + \epsilon_n$, where $\epsilon \sim N(0, \sigma^2 I)$. The low-fidelity likelihood is a function of a numerical solution of the ODE using a time step of size $dt$ (Equation (F.3)). We compared the performance of a multi-fidelity elliptical slice sampler to single-fidelity elliptical slice samplers with step size $dt = 1 \times 10^{-5}, 1 \times 10^{-4}$. For the ODE solver, we considered both an Euler solver and an 4th-order Runge-Kutta solver (Figure F.1). Figure 4a shows 200 trajectories corresponding to parameters sampled from the posterior distributions under each method using the Euler solver. Figure 4b shows posterior mean estimates for each system parameter. We observe that the estimates from the multi-fidelity slice sampler approach the estimates reported by Howard [25] (black dotted lines) in less time than the single-fidelity samplers.

## 5.4 PDE-constrained optimization

We now consider global optimization of a PDE-constrained objective, where an expensive physical simulation is run repeatedly in an outer loop problem. A common approach for global optimization is simulated annealing, which has been applied to constrained global optimization [40]. Consider a model for heat flow in a thin rod of length $L$ with spatial coordinates $x \in [0, L]$. Let $u(x, t)$ represent the temperature in the rod at position $x$ and time $t$, and let $\bar{u}$ represent a desired target temperature. The goal is to minimize a loss function subject to $u$ satisfying a linear heat equation. This objective along with an initial condition and homogenous boundary conditions can be summarized as:

$$
\begin{aligned}
\text{minimize}_u \quad & \|u - \bar{u}\|_2^2 \\
\text{subject to} \quad & \frac{\partial u}{\partial t} = \alpha \cdot \frac{\partial^2 u}{\partial x^2} + 2\beta \cdot u, \\
& u(x, 0) = \sin\left(\pi x / 2\right), \\
& u(0, t) = u(L, t) = 0, \quad x \in [0, L], t \in [0, T],
\end{aligned} \tag{12}
$$

where $\alpha, \beta > 0$ are the system parameters. The goal is to find $\theta = (\alpha, \beta)$ that minimizes the objective and satisfies the constraints. To solve the PDE, we discretize the domain into a grid of size $\Delta x$ and represent the second derivative using the central difference formula. This induces a system of ODEs that we solve numerically using a Tsitouras 5/4 Runge-Kutta method, setting $\Delta t = 0.4\Delta x^2$ so as to satisfy a CFL stability condition. Here the fidelity of the problem is given by the size of the spatial discretization $\Delta x$, which in turn controls $\Delta t$. We compared against two single-fidelity discretizations of the spatial coordinate, where $\Delta x = 5 \times 10^{-3}, 1 \times 10^{-2}$. The results are in Figure 5, where we plot two of the MF results with $\gamma_0 = 0.1, 0.25$. In these examples, the multi-fidelity estimates converge faster than the single-fidelity estimates in wallclock time.

## 5.5 Gaussian process regression parameter inference

Let $X \in \mathbb{R}^{N \times D}$ and consider a Gaussian process regression model with a squared exponential kernel:

$$
f \sim \text{GP}(0, k_\theta), \quad y = f(X) + \epsilon, \quad \epsilon \sim \mathcal{N}(0, \sigma_0^2), \quad k_\theta(x, x') = \exp\left(-\frac{1}{2\theta^2}\|x - x'\|_2^2\right), \tag{13}
$$

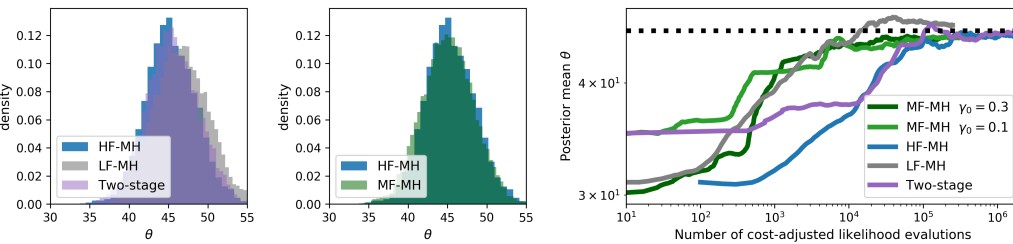

(a) Histograms of high, low, two-stage, and multi-fidelity     (b) Posterior mean estimate vs computation

Figure 6: Parameter inference in a Gaussian process regression model. **Left:** The posterior distribution of the parameter $\theta$. **Right:** The posterior mean estimate vs computional cost.

where we assume $\sigma_0^2$ is known. Let $\Sigma_\theta := [k_\theta(x_i, x_j)]_{i \in [N], j \in [N]}$.

In many applications of Gaussian process modeling, one is interested in integrating out the parameters $\theta$ using MCMC. Computing the posterior $\pi(\theta|X, y)$ is expensive because each evaluation of the likelihood $p(y|X, \theta) = \mathcal{N}(y \,|\, 0, \Sigma_\theta + \sigma_0^2 I)$ involves computing a determinant and solving a linear system with respect to the matrix $\Sigma_\theta + \sigma_0^2 I$, which has an $O(N^3)$ computational cost associated with standard methods (e.g., Cholesky decomposition). For simplicity, we will only consider the fidelity of solution to the linear system, but we note that the determinant can considered using the approach described in Potapczynski et al. [37]. Additional derivations and details are in Appendix F.5.

We generate synthetic data from the GP model with $N = 100$, $\sigma_0^2 = 1$, and lengthscale $\theta_0 = 45$. For the GP model, we use a log Normal prior on $\theta$ given above in Equation (F.5) with parameters $\nu_0 = 3.8, \nu_1 = 0.03$. In Figure 6, we compare these approaches using single-fidelity, multi-fidelity, and two-stage M-H samplers. The low-fidelity likelihood sequence was constructed by computing the solution to the linear system using a preconditioned conjugate gradient solver with $k$ steps. The single-fidelity likelihoods were a high-fidelity likelihood ($k = N$) and a low-fidelity likelihood ($k = k_N \ll N$), and the multi-fidelity M-H samplers used $\gamma_0 = 0.1$. The two-stage M-H approach used high and low fidelities of $k \in \{100, 5\}$. For all methods, we use a M-H sampler with $T = 50000$ iterations. In the histograms, we observe that the high-fidelity, multi-fidelity and two-stage approaches tend to lead to similar posteriors, while the low-fidelity sampler has more noticable bias with respect to the high-fidelity histogram. The estimate produced by the multi-fidelity samplers converged in fewer cost-adjusted likelihood evaluations than the high-fidelity and two-stage approaches.

## 6 Discussion and future work

In this work, we introduced a class of multi-fidelity MCMC that uses a low-fidelity unbiased estimator to reduce the computational cost of sampling while still maintaining the desired limiting target distribution of the Markov chain. In particular, we have demonstrated the use of our framework on more advanced MCMC algorithms beyond M-H, such as slice sampling, and to additional settings such as optimization. Our results show a reduction in computation while producing accurate solutions in comparison with high-fidelity models when it is possible to construct a target estimator that is not too noisy. Many future directions remain. First, applying MF-MCMC to large-scale expensive applications has many computational challenges. Making the method more robust to specialized problems is important, especially if the estimator is heavy-tailed. Thus, constructing good proposal distributions matching properties of the low-fidelity target sequence is crucial, especially for application to high-dimensional problems. In addition, we have thus far focused on target densities where there is a single computation whose fidelity is varied. However, in many settings, there may be target densities with multiple computations that converge at different rates, for example, if the target density includes both an intractable integral and a solution of a linear system. Our framework can be extended to that setting by adjusting the proposal distribution, and it is useful to understand how these rates impact the convergence properties of the sampler.

## Acknowledgments and Disclosure of Funding

This work was partially supported by NSF grants IIS-2007278 and OAC-2118201. D. Cai was supported in part by a Google Ph.D. Fellowship in Machine Learning.

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
