# Multi-fidelity Monte Carlo:
# a pseudo-marginal approach

**Diana Cai**
Department of Computer Science
Princeton University
dcai@cs.princeton.edu

**Ryan P. Adams**
Department of Computer Science
Princeton University
rpa@princeton.edu

## Contents

36th Conference on Neural Information Processing Systems (NeurIPS 2022).

# A    Additional related work

Approximate Bayesian computation (ABC) is a related class of methods to pseudo-marginal MCMC for implicit likelihoods. We note that the asymptotic target for ABC is an approximation to the target density of interest. Prescott and Baker (2020) propose a multi-fidelity approach to ABC. A number of multilevel ABC approaches (Guha and Tan, 2017; Lester, 2018; Warne et al., 2021) have also been proposed in recent work.

In our work, the Russian roulette estimator is used to construct an unbiased, low-fidelity likelihood. The Russian roulette estimator has been used recently in a number of applications for optimization and inference (Beatson and Adams, 2019; Luo et al., 2020; Potapczynski et al., 2021). In particular, Potapczynski et al. (2021) apply similar techniques to get an unbiased estimate of the gradient of the marginal likelihood for Gaussian process regression; this can viewed as an optimization analog to our approach.

# B    Derivation of the sign-corrected estimator

In this section, we provide details for the derivation in Equation (6), and show that the estimator in Equation (5) is asymptotically correct. Recall the definitions of the augmented densities

$$\pi(\theta, K) = \mu(K)\hat{\pi}_K(\theta)$$
$$\tilde{\pi}(\theta, K) = \frac{\mu(K)|\hat{\pi}_K(\theta)|}{\int \sum_{k=1}^{\infty} \mu(k)|\hat{\pi}_k(\theta)|d\theta}.$$

Note that these functions are densities that integrate to 1.

The goal is to derive an unbiased estimator of the functional Equation (1) by rewriting this quantity in terms of the joint target $\tilde{\pi}(\theta, K)$ and then to perform a Monte Carlo estimate with respect to this distribution.

Expanding the functional into the joint distribution, we have

$$\int h(\theta)\pi_\infty(\theta)d\theta = \int h(\theta) \sum_{k=1}^{\infty} \pi(\theta, k)d\theta \tag{B.1}$$

$$= \int \sum_{k=1}^{\infty} h(\theta)\hat{\pi}_k(\theta)\mu(k)d\theta \tag{B.2}$$

$$= \frac{\int \sum_{k=1}^{\infty} h(\theta)\hat{\pi}_k(\theta)\mu(k)d\theta}{\int \sum_{k=1}^{\infty} \hat{\pi}_k(\theta)\mu(k)d\theta}, \tag{B.3}$$

where we applied the equation $\int \sum_{k=1}^{\infty} \mu(k)\hat{\pi}_k(\theta)d\theta = 1$.

Finally, substituting $\hat{\pi}_k(\theta) = \sigma(\theta, k)|\hat{\pi}_k(\theta)|$ in the line above and then multiplying and dividing by the normalizing constant of $\tilde{\pi}(\theta, K)$, we get

$$\int h(\theta)\pi_\infty(\theta)d\theta = \frac{\int \sum_{k=1}^{\infty} h(\theta)\sigma(k, \theta)|\hat{\pi}_k(\theta)|\mu(k)d\theta}{\int \sum_{k=1}^{\infty} \sigma(k, \theta)|\hat{\pi}_k(\theta)|\mu(k)d\theta} \tag{B.4}$$

$$= \frac{\int \sum_{k=1}^{\infty} h(\theta)\sigma(k, \theta)|\hat{\pi}_k(\theta)|\mu(k)d\theta / \int \sum_{k=1}^{\infty} \mu(k)|\hat{\pi}_k(\theta)|d\theta}{\int \sum_{k=1}^{\infty} \sigma(k, \theta)|\hat{\pi}_k(\theta)|\mu(k)d\theta / \int \sum_{k=1}^{\infty} \mu(k)|\hat{\pi}_k(\theta)|d\theta} \tag{B.5}$$

$$= \frac{\int \sum_{k=1}^{\infty} h(\theta)\sigma(\theta, k)\tilde{\pi}(\theta, k)d\theta}{\int \sum_{k=1}^{\infty} \sigma(\theta, k)\tilde{\pi}(\theta, k)d\theta}. \tag{B.6}$$

Now construct a Markov chain with the limiting distribution $\tilde{\pi}(\theta, K)$ and let $\{(\theta^{(t)}, K^{(t)})\}_{t=1}^{T} \sim \tilde{\pi}(\theta, K)$. A Monte Carlo estimate of the previous line then gives the final estimate of the posterior functional:

$$\int h(\theta)\pi_\infty(\theta)d\theta = \frac{\int \sum_{k=1}^{\infty} h(\theta)\sigma(\theta, k)\tilde{\pi}(\theta, k)d\theta}{\int \sum_{k=1}^{\infty} \sigma(\theta, k)\tilde{\pi}(\theta, k)d\theta} \approx \frac{\sum_{t=1}^{T} h(\theta^{(t)})\sigma(\theta^{(t)}, K^{(t)})}{\sum_{t=1}^{T} \sigma(\theta^{(t)}, K^{(t)})}. \tag{B.7}$$

# C   Review of MCMC algorithms

In this section, we review the MCMC algorithms used in the main paper.

## C.1   Random-walk Metropolis-Hastings

Draw proposal $\theta' \sim q(\cdot \,|\, \theta)$. Accept or reject the proposed value according to:

$$R = \min\left(1, \frac{\pi(\theta', \mathcal{D})q(\theta|\theta')}{\pi(\theta, \mathcal{D})q(\theta'|\theta)}\right)$$

In our experiments, we use a proposal distribution of the form $q(\theta'|\theta) = \mathcal{N}(\theta'|\theta, \tau)$, where the parameter $\tau$ needs to be tuned.

## C.2   Slice sampling

Slice sampling (Neal, 2003) is auxillary-variable algorithm that automatically generates proposals without the need for an explicit accept/reject step.

Let $P(\theta)$ denote the joint distribution of interest. Given current state $\theta$, sample a uniform random variable $u \sim \text{Unif}(0, p(\theta))$. This random variable induces a height at the current state given by $u' = uP(\theta)$. A horizontal bracket $(\theta_l, \theta_r)$ is defined around $\theta$ and a proposal $\theta'$ is generated. If $P(\theta') > u'$, the proposal is accepted; otherwise the bracket is decreased. We use the "stepping out" and shrinking procedures for generating and shrinking the proposal bracket, as defined in MacKay (2003, Chapter 29.7).

## C.3   Elliptical slice sampling

Elliptical slice sampling (Murray et al., 2010) is used for inference in latent Gaussian models. Let $f \sim N(0, \Sigma)$ denote the latent $D$-dimesional Gaussian variable of interest, and consider a likelihood $L(f) = p(\mathcal{D} \,|\, f)$. The target distribution of interest is the joint distribution

$$\pi^*(f) = \frac{1}{Z}\mathcal{N}(f; 0, \Sigma)L(f), \tag{C.1}$$

where $Z$ is the marginal likelihood of the model. The algorithm is summarized in Algorithm C.1.

## C.4   Two-stage Metropolis-Hastings

The two-stage MH algorithm assumes a single high fidelity likelihood $L^{\text{HF}}$ and low fidelity likelihood $L^{\text{LF}}$. In each iteration $t$, a proposal $\theta'$ is generated from the proposal distribution $q(\cdot|\theta^{(t-1)})$.

**Stage 1:** The proposal is accepted for the second stage or rejected according to the acceptance probability

$$R^{\text{LF}}(\theta; \theta') = \min\left(1, \frac{\pi(\theta')L^{\text{LF}}(\theta')q(\theta|\theta')}{\pi(\theta)L^{\text{LF}}(\theta)q(\theta'|\theta)}\right), \tag{C.2}$$

where $\theta = \theta^{(t-1)}$. If the proposal is rejected, then the value $\theta^{(t)} = \theta^{(t-1)}$.

**Stage 2:** In the second stage, the proposal $\theta'$ is accepted with probability

$$R^{\text{HF}}(\theta; \theta') = \min\left(1, \frac{\pi(\theta')L^{\text{HF}}(\theta')Q(\theta|\theta')}{\pi(\theta)L^{\text{HF}}(\theta)Q(\theta'|\theta)}\right), \tag{C.3}$$

where the proposal distribution $Q$ satisfies

$$Q(\theta'|\theta) = R(\theta', \theta)q(\theta'|\theta) + \left(1 - \int R(\theta, \theta')q(\theta'|\theta)d\theta'\right)\delta_\theta(\theta'). \tag{C.4}$$

**Algorithm C.1** Summary of the eliptical slice sampling iteration (from Murray et al. (2010, Figure 2)).

1: **Input:** Current state $f$, log-likelihood $L$
2: Choose ellipse $\nu \sim \mathcal{N}(0, \Sigma)$
3: Construct log-likelihood threshold:

$$u \sim \text{Unif}[0, 1]$$
$$\log y = \log L(f) + \log u$$

4: Draw initial proposal, define a bracket

$$\theta \sim \text{Unif}[0, 2\pi]$$
$$[\theta_{\min}, \theta_{\max}] \sim [\theta - 2\pi, \theta]$$

5: Define proposal $f' = f \cos \theta + \nu \sin \theta$
6: **if** $\log L(f') > \log y$ **then**
7:     Accept proposal $f'$
8: **else**
9:     Resize bracket and generate new proposal:
10:     **if** $\theta < 0$ **then**:
11:         $\theta_{\min} = \theta$
12:     **else**:
13:         $\theta_{\max} = \theta$
14:     **end if**
15:     $\theta \sim \text{Unif}[\theta_{\min}, \theta_{\max}]$
16:     Goto Step 5
17: **end if**
18: **Output:** New state $f'$

---

Note that in the algorithm, the integral does not need to be explicitly computed, since if $\theta = \theta'$, the chain remains at the same value, and if $\theta \neq \theta'$, then $Q(\theta'|\theta) = R^{\text{LF}}(\theta; \theta')q(\theta'|\theta)$. Note that the high-fidelity acceptance probability can be easily computed as

$$R^{\text{HF}}(\theta; \theta') = \min\left(1, \frac{L^{\text{HF}}(\theta')L^{\text{LF}}(\theta)}{L^{\text{HF}}(\theta)L^{\text{LF}}(\theta')}\right). \tag{C.5}$$

If the proposal is accepted, then the value $\theta^{(t)} = \theta'$, and otherwise, $\theta^{(t)} = \theta^{(t-1)}$.

# D   Multi-fidelity simulated annealing

In simulated annealing, the goal is to sample from some distribution

$$P(\theta) \propto \exp(-E(\theta)),$$

where $E(\theta)$ is an energy function (the interpretation is, e.g., a negative log-likelihood). If used for optimization, $E(\theta)$ is the function we are interested in minimizing. In the simplest simulated annealing case, we instead sample from the annealed distribution

$$\pi(\theta) \propto \exp(-E(\theta))^{\frac{1}{T}} = \exp(-E(\theta)/T).$$

To adapt this to a multi-fidelity method, we consider energy functions of fidelity $K$, denoted by $E_K(\theta)$. the target densities $\pi(\theta|K)$ and $\pi(K|\theta)$.

To sample from $\pi(\theta|K)$, we accept or reject a proposal $\theta'$ based on:

$$R = \exp\left(-\frac{E_K(\theta') - E_K(\theta)}{T}\right).$$

To sample from $\pi(K|\theta)$, we accept or reject a proposal $K'$ based on:

$$R = \exp\left(-\frac{E_{K'}(\theta) - E_K(\theta)}{T}\right)\left(\frac{\mu(K')}{\mu(K)}\right)^{\frac{1}{T}}.$$

# E    Variance of the sign-corrected estimator

A rough estimate of the variance of the Monte Carlo estimator is (([Lyne et al., 2015])[Appendix B]):

$$\frac{1}{N} \times \left\{ \frac{\sum_{n=1}^{N} \sigma(X_n) h^2(X_n)}{\sum_{n=1}^{N} \sigma(X_n)} - \left( \frac{\sum_{n=1}^{N} \sigma(X_n) h(X_n)}{\sum_{n=1}^{N} \sigma(X_n)} \right)^2 \right\} \times \frac{\hat{V}}{\{1/N \sum_{n=1}^{N} \sigma(X_n)\}^2}, \quad \text{(E.1)}$$

where $\hat{V}$ is an estimate of the common autocorrelation sum.

Note that to control the variance of the MC estimator, it is desirable to minimize the number of negative signs and the variance of the low-fidelity estimator by choosing an appropriate $\mu$ whose tails match that of the low-fidelity density sequence $\pi_k$ ([Beatson and Adams, 2019]; [Potapczynski et al., 2021]).

# F    Experiments: additional experiments and method details

In this section, we provide additional details for the methods used in our experiments along with additional details of the setup of each experiment.

**Methods compared**    We will use the abbreviations *SF* to denote a single-fidelity algorithm, e.g., SF M-H, *MF* to refer to the pseudo-marginal MF-MCMC method proposed in this work, and *TS* to refer to the two-stage M-H algorithm described in Appendix C.4. The primary sampling algorithms used to update the state $\theta | K$ are Metropolis-Hastings (M-H), (line) slice sampling (SS), and elliptical slice sampling (ESS).

**Target estimator $\hat{\pi}$**    In our experiments, by default we consider the Russian roulette estimator with $\mu = \text{geometric}(\gamma_0)$, unless stated otherwise.

**Sampling the fidelity $K|\theta$**    To sample the fidelity from the conditional target $K|\theta$, we consider the following random walk M-H move. Here the target is

$$\pi(K|\theta) \propto \mu(K)\hat{\pi}_K(\theta). \quad \text{(F.1)}$$

To propose a new fidelity, we consider a random walk on the positive integers: flip a fair coin to determine a new candidate location $k^* = k \pm 1$, where $k$ is the current value. Then we can compute the following ratio and decide to accept/reject this candidate value:

$$R = \min \left( 1, \frac{\mu(k^*)\hat{\pi}_{k^*}(\mathcal{D})}{\mu(k)\hat{\pi}_k(\mathcal{D})} \right).$$

In problems where the estimator may return negative values, we compute the absolute value of the estimator $|\hat{\pi}|$, as summarized in Algorithm 1.

## F.1    Toy conjugate sequence

In this example, we consider a toy conjugate Bayesian model, where the data are assumed to arise i.i.d. from a perfect-fidelity model $L_\infty(\theta) = \mathcal{N}(x; \theta, \sigma_\infty)$, and a conjugate prior on $\theta$, $\mathcal{N}(\theta|0, 1)$; conjugacy leads to a closed form Gaussian posterior density that we can compute and compare to the posterior samples obtained from the methods that we compare. Thus, the perfect-fidelity target is $\pi_\infty(\theta) \propto \mathcal{N}(\theta|0, 1) \prod_{n=1}^{N} \mathcal{N}(X_n; \theta, \sigma_\infty)$.

Now suppose that we only have access to the sequence of low-fidelity models $L_k(\theta) = \mathcal{N}(x; \theta, \sigma_k)$, where $\sigma_k^2 \to \sigma_\infty^2$. Here we consider the sequence $\sigma_k^2 = 1 + 2/k^2$ and $\sigma_\infty^2 = 1$. In this example, we consider the performance of (1) SF M-H, MF M-H, and two-stage M-H, and (2) SF slice sampling and slice sampling (there is not an analogous two-stage MCMC algorithm for slice sampling). We generate $N = 200$ observations $\mathcal{D}|\theta_0$ from the perfect-fidelity likelihood with true mean $\theta_0 \sim \mathcal{N}(0, 1)$.

To compute the "cost" of a likelihood evaluation, we pretend that the likelihood evaluation $L_k$ has cost $k$. This is to demonstrate the cost of the method for problems where the cost of an evaluation increases linearly with $k$.

In what follows, we first compare the low-fidelity estimators, and then we compare the sampling methods on one choice of estimator.

**Comparing SF-MCMC, MF-MCMC, and two-stage M-H**  We also compare to the two-stage M-H algorithm summarized in Appendix C.4; here we consider 2 two-stage setups of $k = \{1000, 10\}$ and $k = \{100, 5\}$. For all methods, we ran 4 chains initialized from the prior with $T = 10000$ iterations. We discarded 2000 burn-in samples and the subsequently collected every other sample.

## F.2  Log Gaussian Cox Process

In this section, we provide details for the LGCP experiment on the coal mining disasters data set.

We approximate the integral in Equation (10) with a trapezoidal quadrature rule $I_k$: i.e., given $k$ points $\tilde{x}_1, \ldots, \tilde{x}_k \in \mathbb{X}$ and observed points $\{X_1, \ldots, X_N\}$, the low-fidelity likelihood is:

$$L_k(f) = \exp\left(I_k(f(\tilde{x}_1), \ldots, f(\tilde{x}_K))\right) \prod_{n=1}^{N} e^{f(X_n)}, \tag{F.2}$$

where $I_k$ is a trapezoid quadrature rule with $2k + c$ quadrature points and $c$ is a constant offset parameter. When computing $L_k$ for a grid of values different than the vector of latent function values currently available, we draw new function values conditioned on the existing values of $f$.

For all samplers, we used a squared-exponential kernel with lengthscale $\ell = 20$ and variance of 1. For the low-fidelity estimator $\hat{L}_k$, we used a Russian roulette estimator and set the offset $c = 10$. The truncation parameter of the MF model was fixed at $\gamma_0 = 0.08$. The results in the rightmost figure in Figure 3 are computed with respect to an average over 3 chains initialized from the prior with $T = 10000$ samples. The estimates with MF-ESS in Figure 3 were adjusted for negative signs; empirically, we observed roughly $2.5\%$ of negative signs in our experiments.

## F.3  Bayesian ODE system identification

Given a set of parameters $\theta$ and initial conditions, we can solve the ODE at a fidelity $k$ to obtain the solution $z_n^{(k)}$. Thus, the likelihood of fidelity $k$ is given by:

$$L_k(\theta) = \prod_{n=1}^{N} \prod_{j=1}^{2} \text{LogNormal}(\log(z_{n,j}^{(k)}(\theta)), \sigma), \tag{F.3}$$

where $k$ represents the fidelity of the ODE solver for obtaining the solution $z_n(\theta)$. We use the following priors on the parameters

$$(\log \alpha, \log \beta, \log \gamma, \log \delta) \sim \mathcal{N}(\theta_0, \sigma_0 I), \qquad \theta_0 = [0, -2, 0, -3]^\top, \quad \sigma_0 = 0.1. \tag{F.4}$$

In order to apply elliptical slice sampling, which requires the prior to have mean 0, we apply a change of variables: define $\bar{L}_k(\bar{\theta}) = L_k(\theta + \theta_0)$, and then transform the sampled values $\theta^{(t)} = \bar{\theta}^{(t)} + \theta_0$. In our experiments, we first verified the sampler was recovering values on synthetic data generated with initial conditions $z_0 = [1.0, 1.0]$, system parameters $\alpha = 1.5, \beta = 1.0, \gamma = 3.0, \delta = 1.0$, and noise parameter $\sigma = 0.8$ at a grid of $N$ solution values.

We then applied the method to the Hudson's Bay Lynx-Hare data set, which documents the canadian lynx and showshoe hare populations between 1900 and 1920, based on the data colleﬞted by the Hudson's Bay company. We compared two single-fidelity models with ODE step size $dt = 1 \times 10^{-5}, 1 \times 10^{-4}$. For the multi-fidelity ESS sampler, we visualize the results of $\gamma_0 = 0.12$, and the step size for the low-fidelity target sequence was computed as $dt(k) = 1/(sk + c)$, where we set $s = 10$ and $c = 50$.

The results using Euler's method to solve the ODE are in Figure 4, and the results of the 4th-order Runge Kutta solver are in Figure F.1. The maximum number of iterations of each ODE solver was set to $1 \times 10^8$ iterations.

In the top row of each figure, the black vertical dotted line denotes maximum likelihood estimates reported by Howard (2009).[1] In the bottom row of each figure, we report the posterior mean estimates

---

[1] Our model is a modification of the one proposed in a Stan case study, which compares their Bayesian estimates to the reported maximum likelihood results. See https://mc-stan.org/users/documentation/case-studies/lotka-volterra-predator-prey.html for further discussion.

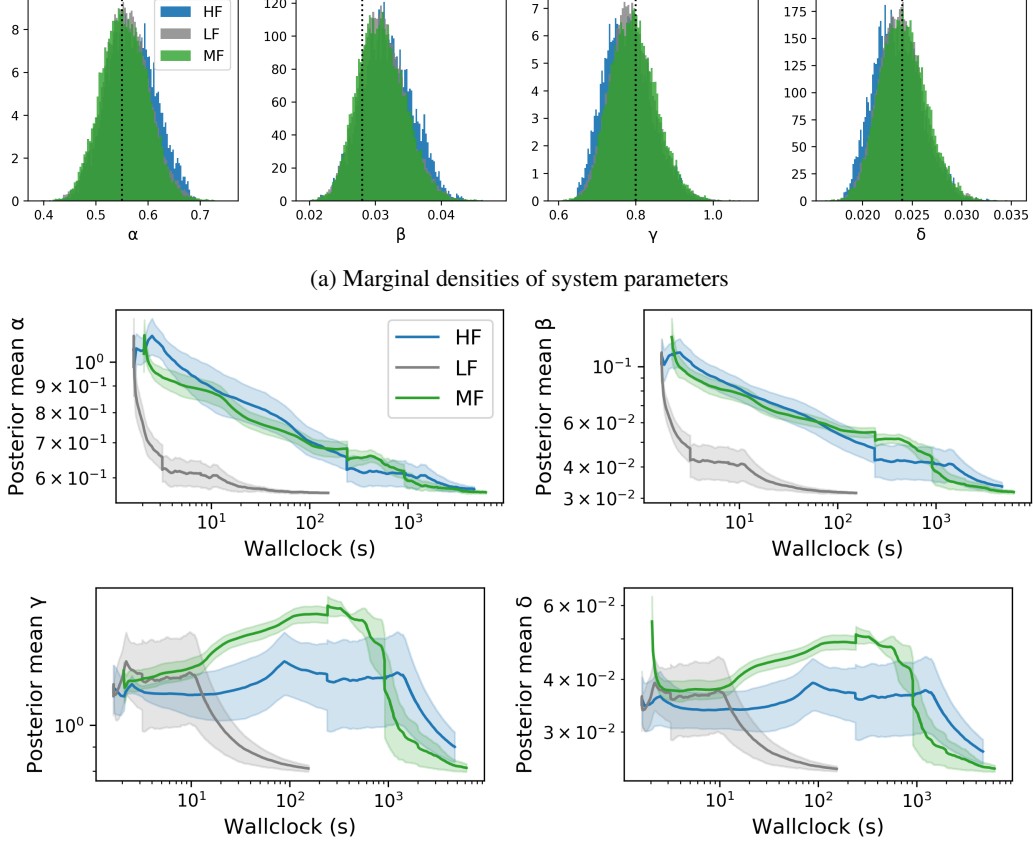

(a) Marginal densities of system parameters

(b) Posterior mean estimate vs computational cost

Figure F.1: Lokta-Volterra system parameter identification with a 4th-order Runge Kutta ODE solver. The fidelity represents (a function of) the step size of the ODE solver. **Top:** Marginal distributions of system parameters. **Bottom:** Posterior mean estimates of the parameters vs wallclock.

of the system parameters averaged over 4 chains initialized from the prior. The wallclock time in seconds of each iteration was measured and the average per iteration was reported. Here the first 5000 samples of each chain were discarded and then every third sample was collected. Overall, we observe that the single-fidelity models can both be quite expensive; while they are able to recover the posterior mean well, they require quite a bit more computation than the multi-fidelity approach. Empirically, we observed roughly $1\%$ of negative signs in our experiments.

### F.4 PDE-constrained optimization

In the problem setting, the spatial domain is $[0, L]$ and the time domain is $[0, T]$. For our experiments, we chose $L = 10$ and $T = 1$.

To solve the PDE, we discretize the spatial domain into a grid of size $\Delta x$: thus, we can consider points $x_1, \ldots, x_I$ and $u_1(t), \ldots, u_I(t)$, where $u_i(t) = u(x_i, t)$. Then, we represent the second derivative using the central difference formula for the second degree derivative:

$$\frac{\partial^2 u(x,t)}{\partial^2 x} \approx \left[ \frac{u_{i+1}(t) - 2u_i(t) + u_{i-1}(t)}{\Delta x^2} \right]_{i=1}^{I}.$$

Thus, we now consider the system of equations (with the appropriate boundary conditions imposed):

$$\frac{u_{i+1}(t) - 2u_i(t) + u_{i-1}(t)}{\Delta x^2} = \frac{du_i(t)}{dt}.$$

We solve the system with the Tsitouras 5/4 Runge-Kutta method, setting $\Delta t = 0.4\Delta x^2$ so as to satisfy a CFL stability condition. Here the fidelity of the problem is given by the size of the spatial discretization $\Delta x$, which in turn controls the discretization of $\Delta t$.

The target temperature $\bar{u}$ was constructed by solving the PDE with parameters $\alpha_0 = 0.85$ and $\beta_0 = 0.21$. For the simulated annealing algorithm, we use a Metropolis-Hastings algorithm as the base sampler; all methods used a truncated Normal proposal with scale set to 0.3 and a logarithmic temperature schedule.

In the top row of Figure 5, we visualization the target $\bar{u}$ solutions recovered by a number of methods. The low-fidelity solution in target (c) is given by a crude step size of $\Delta x = 2$; note that we do not evaluate the cost of this given how poorly the solution is recovered at this state.

In the bottom row of Figure 5, we compare the MF-ESS approach with two single-fidelity step sizes, $\Delta x = 5 \times 10^{-3}, 1 \times 10^{-2}$. In the multi-fidelity method, the low-fidelity target sequence was chosen using the discretization sequence $\Delta x(k) = 1/(k + c)$, where $c = 8$. The results are averaged over random seeds using the initialization $[0, 0]$. The horizontal dotted lines in each plot denote the values of $\alpha_0, \beta_0$, and we plot the current minimum at each iteration.

### F.5 Gaussian process regression parameter inference

In many applications of GPs, the goal is to integrate out the parameters $\theta$ via a Monte Carlo approximation that uses MCMC to sample $\{\theta^{(t)}\}$ from the target density

$$\pi_\infty(\theta \mid \mathcal{D} = (X, y)) \propto \pi(\theta)L_\infty(\theta) = \text{logNormal}(\theta \mid \nu_0, \nu_1) \times \mathcal{N}(y \mid 0, \Sigma_\theta + \sigma_0^2 I). \quad \text{(F.5)}$$

Note that the Gaussian pdf has the form

$$L_\infty(\theta) = |2\pi(\Sigma_\theta + \sigma_0^2 I)|^{-\frac{1}{2}} \exp\left(-\frac{1}{2}y^\top(\Sigma_\theta + \sigma_0^2 I)^{-1}y\right), \quad \text{(F.6)}$$

and so when $N$ is large, the linear system and determinant above become expensive.

Let the low-fidelity likelihood $L_k(\theta)$ denote the computation of the likelihood with $k$ iterations of (preconditioned) conjugate gradient. That is, suppose, $z^{(k)}$ is the $k^{th}$ iteration of the CG with respect to the linear system $(\Sigma_\theta + \sigma_0^2 I)z = y$. Thus, the low-fidelity likelihood is

$$L_k(\theta) = |2\pi(\Sigma_\theta + \sigma_0^2 I)|^{-\frac{1}{2}} \exp\left(-\frac{1}{2}y^\top z^{(k)}\right).$$

In practice, the determinant also needs to be approximated with another low-fidelity computation. Our goal here is to show a proof of concept, and so we only consider the linear system above; however, we note that the determinant can be iteratively computed as a byproduct of conjugate gradient as in Potapczynski et al. (2021). Note that we can compute the likelihood recursively in that each $z^{(k)}$ reuses computation from the previous step $z^{(k-1)}$, and thus a Russian roulette estimator also can reuse computation for each term in the sum.

We generate synthetic data from the GP model with $N = 100$, $\sigma_0^2 = 1$, and lengthscale $\theta_0 = 45$. For the GP model, we use the Log Normal prior on $\theta$ given above in Equation (F.5) with parameters $\nu_0 = 3.8, \nu_1 = 0.03$. We compare several likelihoods: a high-fidelity likelihood ($K = 100$), low-fidelity likelihood ($K = 5$), and the multi-fidelity approach we describe with $\gamma_0 = 0.1$. The low-fidelity likelihood sequence was constructed by computing the solution to the linear system using a conjugate gradient solver with $k$ steps. Finally, we also compare to a two-stage M-H approach with $k \in \{100, 5\}$. For all methods, we use a M-H sampler with $T = 50000$ iterations. The results are in Figure 6.

## G  Broader impacts

In this work, we develop a method that allows for the use of multi-fidelity models in MCMC. However, we do not discuss in detail how to use and interpret the results of MCMC output. Ultimately, any finite sample collected will have initialization bias and generally requires additional choices for the number of burnin samples and the number of samples to thin. These choices are in practice important and should be considered carefully before interpreting or using the resulting posterior estimate.