# OpenReview forum: "Multi-fidelity Monte Carlo: a pseudo-marginal approach"
_NeurIPS.cc/2022/Conference — NeurIPS 2022 Accept_

### Official Review · Reviewer_xJ5d · 2022-06-27

**Rating:** 7
**Confidence:** 4
**Soundness:** 3 good
**Presentation:** 3 good
**Contribution:** 3 good

**Summary:**

“Multi-fidelity Monte Carlo: a pseudo-marginal approach” presents an approach to MCMC that converges faster than many common alternatives. It presents an unbiased estimator for multi-fidelity settings that can be used to construct an MCMC algorithm. The only requirement is that the problem can be written as a sequence of increasing-fidelity target densities.

The authors then show their approach in 5 different experiments on different problems requiring different solution approaches, thereby demonstrating the flexibility of their method. In all cases, the multi-fidelity approach converges much faster than a high-fidelity approach but always finds a solution that is similar in quality to the alternatives. In these experiments, the multi-fidelity approach seems strictly superior to its alternatives.

*UPDATE*: thanks for addressing all of my concerns. I still vote to accept the paper. Furthermore, I feel like the concerns of other reviewers were well addressed or only minor compared to the benefits of the paper and they thus did not reduce my score.


**Questions:**

Please note that I prefer honest answers over answers that prefer the method in the best possible light. If there is a limitation, I’d rather have that stated in the paper and it will not reduce my score.
- Scale: The five experiments you show demonstrate the method can be applied to a host of problems. However, all problems seem like very low-dimensional problems to me (e.g. less than 5D). I would be interested in whether the method also scales to higher-dimensional problems. If the efficiency gains seen in lower dimensions would translate to higher dimensions this would be a major upside of the method and should be worked out. In case it doesn’t scale, it would be interesting to understand why.
- Space of applications: Your method can be applied to all problems that can be phrased as a sequence of increasing-fidelity densities. You already demonstrate that this includes a wide variety of problems in your experiments but I’m not sure how many problems can be phrased like that. Two sentences on which kind of problems this entails would be helpful to me and readers who are new to the field.
- Complexity of application: If I understand correctly, you have to a) find a way to write your problem as a sequence of increasing-fidelity densities, then b) introduce a proposal distribution and c) use equation (3) to compute the expectation. This seems like a very simple procedure that can be applied easily without much new derivation or background knowledge that would only change a few lines of code from a high-fidelity implementation. My question is then: is it really that simple? If that is the case, this could be a major strength of the method and should be emphasized.


**Limitations:**

The paper discusses some limitations implicitly by pointing out the assumptions of the method. It also explicitly points out some limitations in the conclusion and future work section. I think it is good that the authors explicitly state the limitations.

I will use the rest of the section to make some high-level remarks.
- I would increase my score if the reviewers provide good answers to the questions detailed above. More concretely this means:
- a) Provide clarity on whether the method scales to larger settings. In case it doesn’t I would be interested in hypotheses on why it doesn’t.
- b) More clarity on the space of applications. Two sentences listing different possibilities or a general idea of how many problems can be phrased for your method to work are sufficient.
- c) A general idea of the complexity of the method. Very non-technical answers such as “it requires 5 lines of code to change a high-fidelity method to a multi-fidelity method” or “We expect it would take a subject-matter expert X hours to change their algorithm to multi-fidelity” are sufficient.
- I want to help where I can. In case something is unclear, feel free to ask follow-up questions.


**Strengths And Weaknesses:**

*Strengths:*
- The paper is well-written and clear
- The method clearly and strongly outperforms its comparisons on all tested benchmarks
- The method is applicable to a large class of problems

*Weaknesses:* (see questions for details)
- The experiments seem to be run on low-dimensional problems only (correct me if I’m wrong)
- I’m not sure how large the class of problems the method could be applied to is exactly and how easily it is applicable to new problems.

*Typos:*
- 138: fails to
- 143: cite
- 226: and -> an

---

> ### Author Response · Authors · 2022-08-02
> **Response to Reviewer 4**
>
> Thanks for your feedback on the paper and engaging questions. We will work on clarifying the points below in the final version.
>
> **High-dimensional settings:**
>
> High-dimensional problems are indeed challenging, as simple MCMC methods such as M-H will have trouble mixing. Thus, more advanced MCMC are typically needed here, and as we mention in the introduction, it is not obvious how to adapt existing multi-fidelity methods to these more advanced MCMC methods. On the other hand, our MF-MCMC approach makes it straightforward to use multi-fidelity models with more advanced MCMC methods, such as slice sampling.
>
> In our work, one high-dimensional example we studied was in the log Gaussian Cox process (LGCP) experiment. Here the latent (log) rate function is represented by a vector of dimension 191, i.e., the number of observations. For the LGCP, we applied our approach to develop a multi-fidelity elliptical slice sampling algorithm, which makes use of the correlation in the high-dimensional vector. (We added some text in lines 258–259 to make this point more explicit.)
>
> More generally, high-dimensions could lead to robustness challenges due to slow convergence of the MCMC method; in our approach, the additional variance introduced by the randomized low-fidelity estimator would make the convergence more challenging. Thus, the challenge is to make sure the estimator is well-behaved and not too noisy. In the final version, we will expand on our current discussion of challenges in high-dimensions that may occur if the estimator is too noisy.
>
> **Space of problems:**
>
> See our general comment above for a summary. To add to that, we’re motivated by a number of problems that use approximate numerical methods. Multi-fidelity methods are most commonly applied in problems with PDEs, ODEs, and large linear systems, as we study in this paper. Other applications include methods that use optimization as a part of an inner loop; for instance, in Bayesian inverse reinforcement learning, the likelihood depends on the optimal Q-value function, and so an expensive iterative forward RL algorithm needs to be performed at every iteration of MCMC. Another example arises in quantum Monte Carlo simulations, where e.g. approximations for expensive path integrals are used, leading to a biased solution. We’ve also added some more discussion in lines 116–121 of the paper.
>
> **Complexity of implementation:**
>
> Thanks for the suggestion of reiterating the simplicity of the method in the paper. In the revision, we’ve added some text to emphasize the simplicity of implementation in that you just have to apply a standard MCMC algorithm to a low-fidelity model. See, e.g., Lines 162–166 in the updated version.
>
> To answer your question in more detail, the implementation of the method is indeed simple. In a typical vanilla MCMC implementation, you’d implement an iteration function (e.g, slice_iteration) that takes as arguments log target density, the current state (and any hyperparameters) and returns a new state. The multi-fidelity version can use essentially the same function but you’d pass as input to the log target density function a log target *estimator* that depends on a (deterministic) fidelity. To construct the full Markov chain, you’d implement another function that updates the fidelity K; for instance, a simple random walk M-H is a few lines of new code. In particular, the ability to easily apply this muti-fidelity variant to algorithms such as slice sampling leads to new automated multi-fidelity algorithms with minimal tuning parameters; here the main choice we need to make is the truncation distribution. In the final revision, we can add some pseudocode boxes for specific MCMC algorithms that would make Algorithm 1 more concrete to users.
>
> Thanks for catching typos; we’ve corrected those.

---

> > ### Comment · Reviewer_xJ5d · 2022-08-08
> > **Thanks**
> >
> > I updated my review to clarify my stance for the AC. No further action from you is required.
> >
> > UPDATE: thanks for addressing all of my concerns. I still vote to accept the paper. Furthermore, I feel like the concerns of other reviewers were well addressed or only minor compared to the benefits of the paper and they thus did not reduce my score.

---

### Official Review · Reviewer_RpwA · 2022-07-05

**Rating:** 6
**Confidence:** 4
**Soundness:** 4 excellent
**Presentation:** 4 excellent
**Contribution:** 3 good

**Summary:**

This paper proposes a new MCMC algorithm to simulate from a posterior distribution in settings where calculating the likelihood function is computationally expensive. The authors propose a pseudo-marginal based MCMC algorithm which reduces the computational cost of calculating the expensive likelihood by using lower fidelity approximations. The proposed algorithm has the advantage that, via the pseudo-marginal structure, is asymptotically exact in the sense that samples generated from the algorithm will be from the "perfect-fidelity" limiting distribution.

The authors confirm that value of the proposed algorithm through five varied examples and compare their multi-fidelity approach against low and high fidelity alternatives.

**Questions:**

- In the paper the authors state, "*The full multi-fidelity MCMC algorithm with sign correction summarized in Appendix A. We note that while the Markov chain no longer converges to the target, $\pi$  but instead an absolute value target density, the final estimate after sign-correction is unbiased, which is ultimately the downstream goal of interest.*" Can the authors prove this statement?

- In Section 4 the authors state, "*Next, introduce a proposal distribution $\mu$ on the positive integers. This proposal distribution should be chosen to balance between expected cost and variance of the resulting estimator; our overall goal is to mostly use cheap low-fidelity densities, but high variance estimators will presumably damage the mixing time and/or the asymptotic variance.*" It wasn't clear to me how a user should choose this distribution. Could some guidance be given?

**Limitations:**

The authors address some of the limitations of their approach in the Discussion and Future Work section.

The authors only discuss MCMC-type approaches as alternatives to their methodology. I'm not very familiar with the multi-fidelity literature, however, from this paper it appears that a limited range of alternative approaches has been considered. For example, the authors do not discuss Gaussian process-based approaches, such as GP emulation or Bayesian optimisation. Therefore, I think it would be helpful if the authors were able to consider their work in the broader context of this field of research. There's also a related literature on multi-level Monte Carlo which seems to be missing.

**Strengths And Weaknesses:**

Originality - The proposed methodology extends on the more traditional delayed-acceptance type approach to the expensive likelihood problem. The authors utilise a pseudo-marginal scheme which is commonly applied in the literature when dealing with intractable likelihood problems.
Quality - The paper is of a good quality and provides a useful, and theoretically justifiable, solution to the problem of estimating parameters with expensive likelihoods.
Clarity - The paper is very clearly written and the supplementary material provides some additional details which would be helpful to users wishing to use this algorithm. A small area of improvement that the authors could consider would be to include an algorithm box in the main paper to make it a little easier for the reader to understand the algorithm.
Significance - The paper addresses an important problem in the literature, where evaluating expensive likelihood functions for increasingly complex models is a growing issue. However, the experimental section doesn't completely push the boundary on the class of expensive models. The examples used in the paper are nice in the sense that they are simple enough for the reader to understand, but are perhaps not sufficiently complex that a multi-fidelity pseudo-marginal MCMC approach would be essential for inference.

---

> ### Author Response · Authors · 2022-08-02
> **Response to Reviewer 3**
>
> Thank you for your helpful comments on the paper. We’ve made several changes to the manuscript addressing some of your questions, and we will work on clarifying the points below for the final version.
>
> **Sign corrected estimate is unbiased:** We provided a sketch of why this is true at the end of Section 2.2. This result arises from a technique originating from the quantum Monte Carlo literature (see, e.g., Lin et al. (2000) [28]).  Essentially, it results from expanding the target distribution in the functional we’re trying to estimate into its joint distribution (and summing over k) and then substituting with the sign * absolute value of the estimator (see Equation 6 in the revision). The estimate arises from taking a Monte Carlo estimate of the integral and sum (which is an asymptotically unbiased estimator). Here the samples $\theta$ and $K$ arise from simulating from a Markov chain with the target distribution given by the joint distribution $\tilde \pi$. Thus the estimate is asymptotically unbiased. We’ve worked on making this argument more clear in the updated revision, and we added a complete derivation in Appendix B of the revision.
>
> **How to choose the proposal distribution:**
>
> The choice of $\mu$ is well-studied in the literature. For the single-term estimator, $\mu$ has been chosen to minimize variance (see Lyne et al. (2015) [29] and Potapczynksi et al. (2021) [36]) and for the Russian roulette, $\mu$ has been chosen to minimize an objective of the form variance / cost (see Beatson et al. (2019) [5] and Potapczynksi et al. (2021) ). Furthermore Lyne et al. (2015) [29] have noted that in practice a geometric distribution works well. In general, the guidance is to try to match the tails of $\mu$ match with that of sequence of target distributions (as one does with the proposal distribution in importance sampling).
>
> In the updated revision, we’ve added a brief discussion on the truncation distribution $\mu$ in Section 3. In the final version, we can add a section to the appendix with more examples and guidance.
>
> **Related work:** Thanks for the suggestions for improving the related work discussion. We’ll work on expanding the related work discussion for the final version to include more discussion on the broader multi-fidelity literature.
>
> * *Gaussian-process based approaches:* We cited much of this literature in our “general multi-fidelity methods literature” section (see paragraph 2 of the introduction). We’ll note here that this literature is studying a different problem. Typically, they’re interested in reducing the cost of a particular application, such as Bayesian optimization, where $f \sim \text{GP}$ is expensive to evaluate pointwise. On the other hand, we’re really focused on the Monte Carlo problem described in Section 2 and Equation (1). In the final version, we can discuss this connection and the differences in the settings further.
>
> * *Multilevel MC*: We discussed multilevel MC (e.g. Dodwell et al. [12]) in the related work subsection of the introduction (see the first paragraph). We also discussed related multi-fidelity and multilevel approximate Bayesian computation approaches in the additional related work section, which is in the appendix due to space constraints. In our revision, we’ve expanded the multilevel MC discussion in the related work further to make the differences with our method more explicit.
>
> **Clarity:** We agree the algorithm box provides a concise summary of the method for readers and have moved this to the main paper.
>
> **Experiments:** The purpose of our experiments were mostly for understanding the performance of the method. While our models are simple, we note that it is possible to make the existing examples more complex and expensive. For example, in the differential equations, sufficiently fine discretizations become quite expensive to evaluate. For the final version, we intend to run additional experiments for more complex settings. For example, one experiment we have in mind is to generate a more challenging function for the LGCP example, where more quadrature points would be required to get an accurate estimate; more complex functions can also be applied for the differential equation experiments.

---

### Official Review · Reviewer_ga3Z · 2022-07-11

**Rating:** 5
**Confidence:** 3
**Soundness:** 3 good
**Presentation:** 2 fair
**Contribution:** 3 good

**Summary:**

The paper process an alternative approach to what it terms "multi-fidelity" Monte Carlo in which a sequence of estimators of increasing cost and quality (in the sense of approximating some, typically intractable, limiting model) are available. The main innovation of the proposal is to define, explicitly, a target distribution on an augmented space including both a fidelity index parameter and the model parameters of interest. Having done this it is possible to run a Markov chain which targets a distribution over the joint space in order to extract the marginal of interest.

**Questions:**

1: Under what conditions does the proposed approach give rise to estimators of finite variance and with finite expected cost?

As written (5) in the case of single-term estimators (which I specialise to here simply because they simplify the presentation) combined with the sign correction described in line 160 gives a target distribution over k, \theta proportional to

\mu(k) * |\pi_k(\theta) - \pi_{k-1}(\theta)| / \mu_k

whose marginal distribution of k is consequently determined entirely by the normalizing constant of |\pi_k - \pi_{k-1}| and is well defined only if these normalizing constants are summable -- and, indeed, seems independent of the choice of \mu although the situation is more complex in the Russian Roulette setting favoured in the experiments. Is that correct? If so some discussion seems appropriate, the manuscript reads as though \mu is the marginal distribution of K under the target distribution but that seems not to be the case unless I'm misunderstanding.

2: Evaluating (5) seemingly requires that one has access to the normalized "posterior" associated with each fidelity approximation: if one substituted unnormalized quantities in here then the argument doesn't seem to continue to hold. If one took the single term estimator and used unnormalized forms of the individual terms then the weighting described in Q1 would be different, typically, in different terms and so you'd end up with a target that does not have \pi_\infty as a marginal. Is that correct? How is that dealt with in practice?

2: Is the fact that one sees good performance in experiments due to the fact that the target distribution has light enough tails in k to keep the cost under control but heavy enough ones to keep the estimator variance finite in the examples studied, or is it just a consequence of the fact that the Markov chain hasn't run for long enough to start a long excursion into a region of the space for which

3: It would be helpful to state explicitly the target distribution of the Markov chain somewhere prominent in the manuscript, it took me quite some time to (I think) understand what's going on and I think the readership could be expanded somewhat if this was made clearer. That is, the |\hat\pi_k(theta)| which appears in the target might profitably be expanded to illustrate how the complications with normalizing constants are addressed.

Typos etc.:
l99: "that may not even inference problems, such as quantum simulations and PDE-constrained optimization problems." is missing a verb: "be" I think.
l180: "I(N=n)" should this read "I(K=k)"?


**Limitations:**

Appendix G touches on societal implications but given the nature of the paper I'm not sure that it adds much to the manuscript; no more seems required.

Some of my questions hint at what seem to me to be limitations which have not been discussed, but it is possible that I have misunderstood.

**Strengths And Weaknesses:**

Seemingly, the proposed approach addresses some of the issues which arise when implementing methods based around telesoping sums in practice, particularly that it is often impossible to define a distribution over the index parameter which gives rise simultaneously to finite variance and finite expected running cost. There are some details that I haven't found completely transparent but I think I've managed to understand the main ideas.

(2) looks different to the usual description of multi-fidelity models at first glance, but this turns out to be because the $\hat\pi_k$ here is not a probability but rather than estimator of the form in (5) which is signed and will not typically integrate to 1. This is perhaps a little confusing and could be clarified.

---

> ### Author Response · Authors · 2022-08-02
> **Response to Reviewer 2**
>
> Thanks for your helpful comments and careful reading of the manuscript. We will revise the paper to make the following points more clear.
>
> **Explicit statement of the joint target distribution:**
>
> Thanks for the helpful suggestions on more explicitly stating the joint target. In our updated revision, we’ve added Equation (3) that specifies the joint density of interest (in the case where the estimator is nonnegative). Note this quantity recovers the target $\pi_\infty$ as the marginal and integrates to 1. In the lines before and after Equation 5 of the revision, we’ve revised the text to be more explicit about the distributions we’re constructing the target on. In particular, the joint distribution is specified by $\tilde \pi(\theta, K)$ (stated after Equation 5); note that this is defined up to proportionality and the full joint target distribution has an integral $ \int \sum_k|\hat{\pi}_K(\theta)|\mu(K) d\theta$ in the denominator.
>
> **Normalization of the target density:**
>
> The points about the normalization are good questions. Indeed, there is nothing guaranteeing the estimator integrates to 1. From the point of the pseudo-marginal method, we only need to define an augmented joint distribution on an augmented space such that the desired target $\pi_\infty$ is recovered as a marginal. Our treatment here is consistent with that of the pseudo-marginal inference literature; see Murray and Graham (2016) [32] and Lyne et al. (2015) [29]. In our next revision, we will update the text to make this point more transparent.
>
> In the case where we only have access to an unnormalized target function (e.g., Bayesian inference), the estimator is applied to the unnormalized function such that this is unbiased. This is what we do in the Bayesian setting, where we apply the estimator to the likelihood. We want to clarify that we do recover $\pi_\infty$ as the correct marginal in the case where the estimator is nonnegative, which occurs, for instance, if the unnormalized target function is bounded.
>
> For the case where the estimator may return negative values, we do have to apply a trick in order to get the intended estimate; this trick arises from the quantum Monte Carlo literature to deal with the so-called “fermion sign problem” (see, e.g., Lin et al. (2000) [28]) and has been explored by Lyne et al. (2015) [29] for Bayesian inference. The fact that the marginal of Equation 5 (in the revision) recovers the marginal $\pi_\infty$ is used to expand the integral in Equation 1, and this equation is manipulated to rewrite it in terms of the joint target $\tilde\pi$.
>
> As we discuss in the manuscript, the joint target is now $\tilde\pi(\theta, K) \propto |\hat{\pi}_K(\theta)|\mu(K)$ which does not have $\pi_\infty$ as the marginal (this is true whether or not we have access to the normalized target density – see Appendix B). But we can still recover the desired posterior functional and get a variety of desired estimates such as posterior means and even histograms used to visualize marginal distributions. (See the discussion at the end of Section 2 for details.) In the unnormalized case, we can more explicitly express this joint in terms of the unnormalized function; for example, if $\pi_0$ is a prior and $\hat{L}_K$ is the likelihood estimator, the joint target is then $\tilde\pi(\theta, K) \propto \pi_0(\theta) |\hat{L}_K(\theta)|\mu(K)$.
>
> We will work on clarifying these points in the final revision of the manuscript. We are happy to include a version of the derivations for the unnormalized case if this helps to improve the clarity of presentation.
>
> **Properties of the estimator and performance:**
>
> There are several existing papers that study when these estimators lead to finite variance with finite expected cost; an overview is given in Lyne et al. 2015 [29]. For example, in the single-term estimator, when the sequence is a geometric series, using the geometric distribution as the proposal distribution leads to finite variance subject to a condition on the parameter; a similar condition establishes finite variance for the Russian roulette estimator.
>
> Regarding performance under heavy tails of the estimator: convergence in situations where the estimator has heavy tails is indeed challenging. This is a situation where engineering the proposal distribution $\mu$ such that it matches the tails of the sequence becomes important. We will expand our current discussion in Sections 3 and 6 to make this point more explicit.
>
> Thanks for catching typos; we’ve fixed these in the updated version.

---

> > ### Comment · Reviewer_ga3Z · 2022-08-07
> > **Response**
> >
> > Many thanks for the clarifications. I think you're doing what I thought but it's good to have concrete clarification.
> >
> > I do have a concern about the use of a "mixture of unnormalized densities" target. This is exactly the situation in simulated annealing and its known that it's important to have mixture weights which compensate for the differences in normalization in order to obtain tolerable mixing. Combined with the difficulties that unbiased methods based around Rhee-Glynn type ideas have with obtaining unbiasedness concurrently with finite expected running time I do worry that this algorithm is likely to share these issues in realistic settings -- at least without theoretical arguments to show that it does not.

---

### Official Review · Reviewer_G3Z4 · 2022-07-15

**Rating:** 6
**Confidence:** 3
**Soundness:** 3 good
**Presentation:** 3 good
**Contribution:** 3 good

**Summary:**

This paper introduces a muti-fidelity MCMC algorithm for the setting where a sequence of likelihoods with increasing fidelity are available. The authors use pseudo-marginal MCMC and construct an unbiased estimation for high-fidelity likelihood by randomized truncation of a telescoping series. In theory, the proposed method can converge to the target distribution asymptotically. In practice, several tasks including log-Gaussian Cox process, Bayesian ODE system identification, PDE-constrained optimization, and Gaussian process regression, are conducted to demonstrate the effectiveness of the proposed method.

**Questions:**

In the experiments, the fidelity k was set to be 10 (which is very low) and 1000 (which is very high). I wonder what if we use a proper k, e.g. we can treat k as a hyperparameter and tune it. How is the performance of such single fidelity compared to multi fidelity? My guess is that this single fidelity can also balance the trade-off between bias and variance. How is K in the two-stage methods chosen? I wonder if two-stage methods can perform as well as the proposed multi-fidelity method by tuning hyperparameters.

It might be better to add a comparison in terms of running time for all tasks. The number of likelihood evaluations may not be a proxy for cost since the cost of evaluating the likelihood in different methods is quite different.

By looking at the empirical results, the proposed method does not seem to improve the results significantly. Specifically, in Sections 5.1, 5.2, 5.3, and 5.5, the proposed method all performs similarly to previous methods. Only in Figure 5 PDE-constrained optimization, the improvement over baselines seems significant, though it is not clear to which baseline “Delta x” refers to.


**Limitations:**

The authors have discussion limitations in the paper.

**Strengths And Weaknesses:**

Pros
- The problem setting is reasonable, that is, assuming a sequence of likelihoods with increasing fidelity are available.

- As far as I understand, the proposed method is technically sound. One nice property of the method is that it is asymptotically correct whereas previous multi-fidelity methods are not in the setting of an infinite number of increasing likelihoods.

- Five different experiments are conducted to demonstrate the proposed method.

Cons:
- The empirical improvement is incremental.

- The setup of baseline methods is unclear in the experiments.

In summary, I think the main weakness of this paper is the weak empirical performance of the proposed method. If the authors can demonstrate its strong empirical performance over previous methods, I will consider improving the score.

---

> ### Author Response · Authors · 2022-08-02
> **Response to Reviewer 1**
>
> Thanks for your helpful feedback on the paper. We’d like to clarify a few of the points, answer questions from the review, and also outline what changes we’ll make to the final version to incorporate your feedback.
>
> **On the fidelity for single-fidelity and two-stage methods:**
>
> This is a good question that’s worth expanding on further. Even with the “optimal” values for the fidelity w.r.t. bias vs compute, the single-fidelity and two-stage methods have an irreducible bias. For the single-fidelity and two-stage methods: it is difficult to tune the fidelity because in most problems, the bias of the estimator is unknown. Our approach can actually be viewed as making a more “automated” multi-fidelity method in that we don’t have to make choices about K directly, and in our case, the error arises from Monte Carlo error and is therefore reducible by taking more samples.
>
> **Baselines:**
>
> The purpose of the single-fidelity baselines was to select a high-fidelity model, where we expect to get very accurate results, and a low-fidelity results, where the results are very cheap to compute but have some noticeable bias; this is designed to emulate scenarios that these methods were designed for. For the LGCP example, K=1000 was chosen for the high-fidelity model since we expect this estimate to be close to the limiting value, and K=10 for the low-fidelity model. Other values were chosen depending on the application. In the PDE-constrained optimization example, “Delta x” refers to a single fidelity with a single fixed spatial discretization of size $\Delta x$. We’ll clarify this in the text to be more explicit, and add more details of all the single-fidelity baselines in the text.
>
> **Empirical performance of methods:**
>
> The main purpose of the experiments was to understand how the MF-MCMC approach performs for models we understand without worrying about differences in implementation details of e.g. numerical methods. The purpose of different experiments also varied depending on whether or not the computation was expensive to evaluate (e.g., the conjugate Gaussian example is cheap to evaluate whereas the differential equations are more expensive to solve). We note that for Bayesian inference experiments, it is common to measure cost-adjusted performance in terms of number of likelihood evaluations, e.g., see Murray and Graham (2016) [32].
>
> In our experiments, we show that the multi-fidelity model is able to get to similar or better solutions in less time than the high-fidelity models. In many cases, we see the cheap, low-fidelity model converging faster but to a more biased estimate.
>
> Furthermore, the estimate vs computation plots currently only compute the posterior mean. Thus, bias in the full posterior may not be as obvious in those plots as those that visualize the inferred posterior. If the goal is to compute accurate estimates of other types of posterior functions, e.g., a variance or measure of risk that depends on a quantile, then the bias in the full posterior will be more visible. This behavior also occurs in the LGCP and Lokta-Volterra experiments; here the mean estimates of the multi-fidelity and low-fidelity models are quite similar but the estimates of the standard deviation are quite different (on the other hand, the multi-fidelity method matches the high-fidelity standard error estimates well). Thus, in some cases we expect to see better performance if we plot other functional estimates besides just the posterior mean.
>
> Finally, as we discuss in the introduction, it’s not obvious how to extend two-stage methods to more advanced methods such as slice sampling; thus, the proposed MF-MCMC approach has the advantage of being applied to other methods such as slice sampling. This is an important step towards developing more automated multi-fidelity MCMC methods that can be applied in higher-dimensional problems.
>
> In the final version, we’ll revise the text to make the interpretation of the results more explicit to the reader. In addition, we will work on improving the experiments to make the gains in results more explicit (e.g., plot other posterior functions besides the mean, consider higher-dimensional examples and more challenging synthetic data scenarios); see Appendix F.1.1 for a preliminary version of this plot.

---

> > ### Comment · Reviewer_G3Z4 · 2022-08-08
> > **Thanks**
> >
> > I thank the authors for their responses. However, the question of the comparison with previous methods with tuned K remains. I agree that the error of the proposed method can be reduced by using large MC samples, but in practice, we could not use an infinite number of MC samples and the number is also a hyperparameter to be tuned, just as K. I think it is a fairer comparison for baselines to have appropriate hyperparameters, instead using extremely large or small values of K.
> >
> > It is also common to measure cost in Bayesian inference, e.g [1]. Given the cost per iteration is very different for different methods, I believe it is worth adding a performance comparison w.r.t. the overall cost.
> >
> > I agree that showing other metrics rather than the posterior mean might reflect the estimation results better. I encourage the authors to add these results in the revision.
> >
> > [1] Scalable Metropolis–Hastings for Exact Bayesian Inference with Large Datasets, ICML 2019

---

> > > ### Author Response · Authors · 2022-08-09
> > > **Thanks for the response**
> > >
> > > Thanks for your response and further engagement in the discussion!
> > >
> > > **Tuning the fidelity K:**
> > >
> > > In practice, there’s two types of biases: (1) the bias from a finite number of samples (which both methods will have), and (2) asymptotic bias (the fixed K methods will suffer from, whereas our method does not). For the first scenario, it’s possible to do diagnostic checks, e.g., starting the Markov chain from many different initial states to get a sense of convergence of the Markov chain, where a properly mixed samples should "forget" the initial position.
> > >
> > > However, in the case of asymptotic bias, in practice we don’t know how much bias there is and so it is not clear how to tune K. After performing the usual diagnostic checks for convergence of the sampler, simulating from the limiting distribution leads to a biased answer. As we mentioned in our general comment, there are many applications where scientists are really interested in accuracy of the method, while also reducing cost, and so the additional asymptotic bias is undesirable. In addition, it is often preferable to not have additional tuning parameters.
> > >
> > > We agree these tradeoffs are worth more discussion in the paper and will make this point more clear. In addition, the $K \in \\{10,1000\\}$ example describes one specific experiment where there is no ground truth (the LGCP); the largest value is chosen by design as a close approximation to the truth. We'd be happy to add a larger grid of values for the fidelity in the next revision. As we mentioned in the previous response, we'll clarify baselines and report other comparison metrics (e.g., beyond the posterior mean).
> > >
> > > **Measuring cost:**
> > >
> > > Thanks for the reference to the paper. We agree with measuring the total cost: to be clear, our experiments are not measuring the quality of the estimate per likelihood evaluation but rather w.r.t. a total cumulative cost. Here each likelihood-evaluation is upweighted by the cost of its evaluation, where we have made the cost linear in the fidelity. We will make this point more clear in the paper.

---

### Author Response · Authors · 2022-08-02
**General response to all reviewers**

We thank the reviewers for their time and thorough engagement with the paper. We appreciate that reviewers overall found the work well-written and that it addresses an important problem. Below, we’ll summarize a few main points relevant to all reviewers and then we’ll also provide a summary of the changes in the updated revision.

**Review of contributions:**

We develop a class of multi-fidelity MCMC algorithms that produces asymptotically unbiased estimates of functionals with respect to an expensive target density. In particular, we propose the first MCMC algorithm that provably estimates functionals in the infinite-fidelity limit without bias from finite truncation. Furthermore, the method is simple to implement, and it is easy to modify an existing MCMC algorithm to produce a multi-fidelity MCMC method. This has the advantage that it can be applied to more advanced MCMC methods in a straightforward manner, in contrast to existing multi-fidelity MCMC methods that are really specialized for Metropolis-Hastings (M-H), thus limiting their utility.

The standard approach is to implement MCMC with respect to a single high-fidelity model; this is computationally expensive and also asymptotically biased with respect to the “perfect-fidelity” target density of interest. Existing multi-fidelity approaches successfully reduce the computational cost w.r.t. high-fidelity models by incorporating low-fidelity models, but these approaches are also biased (in addition to being specialized to M-H).

We see our method being impactful in applications where getting accurate solutions and uncertainties is crucial, e.g., in computational chemistry where the goal is to accurately predict the outcome of an experiment, and using a method that is both expensive and asymptotically biased is undesirable.

**Experiments:**

Our intention with the experiments were to showcase proof-of-concept examples with (1) a pedagogical conjugate Gaussian example with exact ground truth to demonstrate the method works as intended and (2) a variety of different computation sequences that arise in real applications, including quadrature estimates, time discretizations in ODEs, spatial discretizations in PDEs, iterative updates arising from a linear solver.

In the experiments, we demonstrate that our method is often able to converge to better estimates with less computation than the high-fidelity and two-stage methods baselines. We note that it is only straightforward to apply the two-stage baseline to the M-H algorithm, and so we don’t implement it for examples where it is standard to use more advanced MCMC algorithms.

In addition, both baselines are asymptotically biased and so there’s no way to reduce the bias by simply drawing more samples, as we can do in our case. This is most useful in applications where the MCMC user truly cares about the truth w.r.t. the perfect fidelity model (e.g., in applications such as computational chemistry where scientists are happy to draw more samples if it can accurately simulate an experiment because it’s still much less expensive than synthesizing the actual experiment in a lab).

In the final revision, we will make the intended purpose of each experiment and setup of the baselines more clear to the reader.

**Summary of changes in the uploaded revision:**
* Revised Section 2 to be more explicit about (joint) target distributions, signed quantities, and space of applications
* Added a detailed derivation of the sign-corrected estimator to Appendix B
* Added a discussion on the truncation distribution $\mu$ in Section 3
* Moved Algorithm A.1 from the appendix to Section 4
* Moved details about baselines from appendix to Section 5

To make space for these changes, we’ve moved some figures to the appendix. However, for the final revision, we will work on reformatting the manuscript to include more of the most insightful discussion and experiments in the main document and incorporate any remaining feedback from reviewers.

---

### Meta-Review · Area_Chair_D8MC · 2022-08-29

**Recommendation:** Accept
**Confidence:** Certain

**Metareview:**

In "multi-fidelity" Monte Carlo, a sequence of estimators of increasing cost and quality (in the sense of approximating some, typically intractable, limiting model) are available. The authors define a target distribution on an extended space including both fidelity and model parameters. Then, a Markov chain which targets a distribution over this extended space can recover the marginal of interest.

All reviewers were positive about the work and there seems to be a consensus that the paper is well-written and clear, the method is applicable in a broad setting and demonstrates an advantage in tested benchmarks (albeit perhaps low dimensional),
Reviewer ga3Z raised a valid concern about the mixing time considering the nature of the unbiased estimation methods based on couplings described in section 3. These methods achieve unbiased estimates at the expense of estimation variance and expected running time (time to coupling). The reviewer mentions that this algorithm can still suffer from the same problems that annealing based methods can suffer in terms of mixing. Such limitations should be made explicit in the paper.

After the rebuttal, I feel that most concerns have been addressed and the paper can be a valuable contribution to Neurips, hence I suggest acceptance.


**Award:**

No

---

### Decision · Program_Chairs · 2022-09-14

Accept